# The genomic loci of specific human tRNA genes exhibit ageing-related DNA hypermethylation

Richard J. Acton[1,2,3], Wei Yuan[4,5], Fei Gao [6], Yudong Xia[6], Emma Bourne[7], Eva Wozniak[7], Jordana Bell [4], Karen Lillycrop[3], Jun Wang[8,9,10], Elaine Dennison[2], Nicholas C. Harvey [2], Charles A. Mein[7], Tim D. Spector [4], Pirro G. Hysi [4], Cyrus Cooper [2] & Christopher G. Bell [1✉]

The epigenome has been shown to deteriorate with age, potentially impacting on ageing-related disease. tRNA, while arising from only ~46 kb (<0.002% genome), is the second most abundant cellular transcript. tRNAs also control metabolic processes known to affect ageing, through core translational and additional regulatory roles. Here, we interrogate the DNA methylation state of the genomic loci of human tRNA. We identify a genomic enrichment for age-related DNA hypermethylation at tRNA loci. Analysis in 4,350 MeDIP-seq peripheral-blood DNA methylomes (16–82 years), identifies 44 and 21 hypermethylating specific tRNAs at study-and genome-wide significance, respectively, contrasting with none hypomethylating. Validation and replication (450k array and independent targeted Bisuphite-sequencing) supported the hypermethylation of this functional unit. Tissue-specificity is a significant driver, although the strongest consistent signals, also independent of major cell-type change, occur in tRNA-iMet-CAT-1-4 and tRNA-Ser-AGA-2-6. This study presents a comprehensive evaluation of the genomic DNA methylation state of human tRNA genes and reveals a discreet hypermethylation with advancing age.

[1] William Harvey Research Institute, Barts & The London School of Medicine and Dentistry, Charterhouse Square, Queen Mary University of London, London, UK. [2] MRC Lifecourse Epidemiology Unit, University of Southampton, Southampton, UK. [3] Human Development and Health, Institute of Developmental Sciences, University of Southampton, Southampton, UK. [4] Department of Twin Research & Genetic Epidemiology, St Thomas Hospital, King's College London, London, UK. [5] Institute of Cancer Research, Sutton, UK. [6] BGI-Shenzhen, Shenzhen, China. [7] Barts & The London Genome Centre, Blizard Institute, Barts & The London School of Medicine and Dentistry, Queen Mary University of London, London, UK. [8] Shenzhen Digital Life Institute, Shenzhen, Guangdong, China. [9] iCarbonX, Zhuhai, Guangdong, China. [10] State Key Laboratory of Quality Research in Chinese Medicine, Macau University of Science and Technology, Taipa, Macau, China. ✉email: c.bell@qmul.ac.uk

Ageing is implicated as a risk factor in multiple chronic diseases[1]. Understanding how the ageing process leads to deteriorating biological function is now a major research focus, with hopes to increase the human 'healthspan' and ameliorate the extensive physical, social and economic costs of these ageing-related disorders[2]. Epigenetic processes, which influence or can inform us about cell-type specific gene expression, are altered with age and are, furthermore, one of the fundamental hallmarks of this progression[3,4].

DNA methylation (DNAm) is the most common epigenetic modification of DNA and age-associated changes in this mark in mammalian tissues have been recognised for decades[5]. In fact, these alterations in DNAm with age are extensive with thousands of loci across the genome affected. Many represent 'drift' arising from the imperfect maintenance of the epigenetic state[6]. However, specific genomic regions show distinct directional changes, with the loss of DNA methylation in repetitive or transposable elements[7], as well as gains in certain promoters, including the targets of polycomb repressor complex[8] and bivalent domains[9]. These observations with the advent of high-throughput DNAm arrays also permitted the identification of individual CpG sites that exhibit consistent changes with age, enabling the construction of predictors of chronological age known as epigenetic or DNAm 'clocks'[10–13]. In addition, it was observed that 'acceleration' of this DNAm-derived measure is a biomarker of 'biological' ageing due to associations with morbidity and mortality (Reviewed in[14,15]). In a previous investigation of ageing-related DNAm changes within common disease-associated GWAS regions, we identified hypermethylation of the specific transfer RNA gene, tRNA-iMet-CAT-1-4[16]. The initiator methionine tRNA possesses certain unique properties, including its capacity to be rate limiting for translation[17], association with the translation initiation factor eIF2[18], and ability to impact the expression of other tRNA genes[19].

tRNAs are evolutionarily ancient[20] and fundamental in the translation process for all domains of life. This translation machinery and the regulation of protein synthesis are controlled by conserved signalling pathways shown to be modifiable in longevity and ageing interventions[21]. In addition, beyond their core role in the information flow from DNA to protein sequence, tRNAs can fragment into numerous tRNA-derived small RNAs (tsRNAs)[22] with signalling and regulatory functions[23–26].

When liberally considering the 610 loci annotated as tRNA genes, as well as tRNA pseudogenes, nuclear-encoded mitochondrial tRNA genes and possibly some closely related repetitive sequences, these features (gtRNAdb)[27] cover <46 kb (including introns) which represents <0.002% of the human genome[28]. Yet a subset of these genes produce the second most abundant RNA species next to ribosomal RNA[29] and are required for the production of all proteins. tRNA genes are transcribed by RNA polymerase III (polIII)[30] and have internal type II polIII promoters[31]. DNAm is able to repress the expression of tRNA genes experimentally[32] but may also represent co-ordination with the local repressive chromatin state[33]. Transcription is also repressed by the highly conserved polIII specific transcription factor Maf1[34,35], whose activity is modulated by the Target of Rapamycin Kinase Complex 1 (TORC1)[36]. TORC1 is a highly conserved hub for signals that modulate ageing[37].

tRNAs as well as tsRNAs are integral to the regulation of protein synthesis and stress response, two processes known to be major modulators of ageing. Metabolic processes are also recognised to modulate the age estimates of DNAm clocks[38]. Partial inhibition of translation increases lifespan in multiple model organisms[39] and PolIII inhibition increases longevity acting downstream of TORC1[40]. Furthermore, certain tsRNAs circulating in serum can be modulated by ageing and caloric restriction[41].

We directly investigated ageing-related changes in the epigenetic DNA methylation state of the entire tRNA gene set, facilitated by the availability of a large-scale MeDIP-seq dataset. Arrays poorly cover this portion of genome, with even the latest EPIC (850k) arrays only covering <15% of the tRNA genes, with robust probes, and in total only ~4.7% of all the tRNA gene CpGs[42]. tRNA genes sit at the heart not only of the core biological process of translation but at a nexus of signalling networks operating in several different paradigms, from small RNA signalling to large-scale chromatin organisation[43]. In summary, tRNA biology, protein synthesis, nutrient sensing, stress response, and ageing are all intimately interlinked. In this study, we have identified tRNA gene DNA hypermethylation and independently replicated this newly described ageing-related observation.

## Results

**DNA methylation of specific tRNA gene loci changes with age.** Due to tRNAs critical role in translation and evidence of their modulation in ageing and longevity-related pathways, we interrogated these genes for evidence of ageing-related epigenetic changes. Our discovery set was a large-scale peripheral blood-derived DNA methylome dataset comprising of 4350 samples. This sequencing-based dataset had been generated by Methylated DNA Immunoprecipitation (MeDIP-seq)[44], which relies on the enrichment of methylated fragments of 200–500 bp to give a regional DNAm assessment (500 bp semi-overlapping windows). There are 416 high-confidence tRNA genes in the human genome, we initially examined an expanded set of 610 tRNA-like features identified by tRNAscan including pseudogenes, possible repetitive elements, and nuclear-encoded mitochondrial tRNAs[27,45]. Of these 492 were autosomal and did not reside in blacklisted regions of the genome[46]. Due to the small size of these tRNAs (60–86 bp, median 73 bp, excluding introns which are present in ~30 tRNAs with sizes from 10–99 bp, median 19 bp), this fragment-based method enabled a robust examination of the epigenetic state of these highly similar sequences. This was supported by a mappability assessment. The median mappability score density for all tRNA genes was 0.90 for 50mers when considering tRNA genes ± 500 bp reflecting the regional nature of the MeDIP-seq assay. In contrast, the 50mer mappability density is 0.68 for tRNA genes alone, which would be representative of the mappability of reads generated using a technique such as whole-genome bisulfite (BiS) sequencing (see Supplementary Figs. 1 and 2).

We identified 21 genome-wide significant and 44 study-wide significant results ($p < 4.34 \times 10^{-9}$ and $8.36 \times 10^{-5}$, respectively), via linear regression ($n = 4350$). Study-wide significance was calculated conservatively with the Bonferroni method for all 598 autosomal tRNAs and closely related features. There was a strong directional trend with all results at both significance levels being due to increases in DNA methylation. Age-related changes in cell-type proportion are strong in heterogeneous peripheral blood and include a myeloid skew, loss of naive T cells and increases in senescent cells[47]. A subset of 3 genome-wide and 16 study-wide significant hypermethylation results remained significant even after correcting for potential cell-type changes by including lymphocytes, monocytes, neutrophils and eosinophil cell-count data ($n = 3001$, Listed in Fig. 1, see Supplementary Fig. 3). Going forward we refer to these cell-type corrected sets of 3 and 16 tRNA genes as the genome-wide and study-wide significant tRNA genes, respectively.

Due to the related nature of these twin samples, we also analysed these data in two subsets of $n = 1198$ and 1206 by selecting one twin from each pair into the separate sets. This analysis also included correction for blood cell counts. Even in

| tRNA | Window | MeDIP | | 450k Array | | Targeted BiS-seq | |
|---|---|---|---|---|---|---|---|
| | | Slope | p-value | Slope | p-value | Slope | p-value |
| tRNA-Gln-CTG-7-1 | Chr1:147,800,750-147,801,250 | 0.84 | 2.60e-05 | | | | |
| tRNA-Glu-TTC-1-1 | Chr2:131,094,500-131,095,000 | 1.11 | 4.64e-09 | | | | |
| | Chr2:131,094,250-131,094,750 | 1.00 | 1.12e-07 | | | | |
| | Chr2:131,094,750-131,095,250 | 1.00 | 3.28e-07 | | | | |
| tRNA-His-GTG-1-2 | Chr1:146,544,500-146,545,000 | 0.92 | 1.38e-06 | | | | |
| tRNA-His-GTG-2-1 | Chr1:149,155,750-149,156,250 | 1.05 | 2.98e-08 | | | | |
| | Chr1:149,155,500-149,156,000 | 0.83 | 1.37e-05 | | | | |
| tRNA-Ile-AAT-10-1 | Chr6: 27,251,500- 27,252,000 | 1.07 | 1.45e-08 | | | 1.30 | 1.22e-03 |
| | Chr6: 27,251,750- 27,252,000 | 0.90 | 1.86e-06 | | | 1.30 | 1.22e-03 |
| tRNA-Ile-AAT-4-1 | Chr17: 8,130,000- 8,130,500 | 1.19 | 2.98e-10 | 0.20 | 8.92e-06 | -0.74 | 6.88e-04 |
| | Chr17: 8,130,250- 8,130,750 | 0.77 | 3.99e-05 | 0.20 | 8.92e-06 | -0.74 | 6.88e-04 |
| tRNA-Ile-TAT-2-2 | Chr6: 26,987,750- 26,988,250 | 0.97 | 7.25e-07 | 0.04 | 1.17e-02 | -0.60 | 3.84e-01 |
| tRNA-iMet-CAT-1-4 | Chr6: 26,330,500- 26,331,000 | 1.28 | 2.83e-11 | 0.13 | 6.07e-06 | 4.54 | 9.35e-04 |
| | Chr6: 26,330,250- 26,330,750 | 1.13 | 2.89e-09 | 0.13 | 6.07e-06 | 4.54 | 9.35e-04 |
| tRNA-Leu-TAG-2-1 | Chr14: 21,093,250- 21,093,750 | 1.04 | 9.38e-08 | | | 2.49 | 8.77e-03 |
| | Chr14: 21,093,500- 21,094,000 | 0.94 | 8.50e-07 | | | 2.49 | 8.77e-03 |
| tRNA-Pro-AGG-2-2 | Chr6: 26,555,500- 26,556,000 | 1.04 | 3.97e-08 | | | | |
| | Chr6: 26,555,250- 26,555,750 | 1.01 | 9.58e-08 | | | | |
| tRNA-Ser-ACT-1-1 | Chr6: 27,261,250- 27,261,750 | 0.97 | 3.53e-07 | | | 0.66 | 1.45e-01 |
| tRNA-Ser-AGA-2-6 | Chr17: 8,129,750- 8,130,250 | 1.21 | 1.16e-10 | 0.21 | 6.72e-05 | 0.62 | 4.28e-02 |
| | Chr17: 8,130,000- 8,130,500 | 1.19 | 3.03e-10 | 0.21 | 6.72e-05 | 0.62 | 4.28e-02 |
| tRNA-Ser-TGA-2-1 | Chr6: 27,513,000- 27,513,500 | 0.90 | 3.58e-06 | 0.87 | 1.38e-04 | -0.25 | 5.74e-01 |
| tRNA-Val-AAC-1-2 | Chr5:180,590,750-180,591,250 | 0.91 | 3.28e-06 | | | | |
| tRNA-Val-AAC-4-1 | Chr6: 27,648,500- 27,649,000 | 1.07 | 1.25e-08 | 0.40 | 9.90e-03 | | |
| | Chr6: 27,648,750- 27,649,250 | 0.95 | 4.31e-07 | 0.40 | 9.90e-03 | | |
| tRNA-Val-CAC-2-1 | Chr6: 27,247,750- 27,248,250 | 0.85 | 2.33e-05 | 0.59 | 5.05e-06 | | |

**Fig. 1 Ageing-related DNA methylation changes in tRNA loci (16 Study-wide significant after blood cell-type correction, $p < 8.36 \times 10^{-5}$).** Columns left to right: tRNA gene; Genomic location of MeDIP-seq significant window (hg19); MeDIP results slope (arbitrary units) and $p$ value; 450k Array results slope (beta) and $p$ value (Blood-cell corrected and combined for all probes overlapping tRNA); Targeted Bisulphite-sequencing (BiS-seq) slope (model beta) and $p$ value (combined all CpGs in targeted region). Units of slope are quantile normalised RPM values or MeDIP-seq models and proportion of CpG sites methylated for the targeted bisulfite and array based assays, all per unit time. The result for all covered tRNAs are included in the Supplementary data 1, 3, and 4. All $p$ values are for F-tests from simple linear regression. For slope: orange = hypermethylation and blue = hypomethylation with age. For $p$ values: Colour gradient high significance = dark blue to low significance = yellow, scaled relative to column. Blank grey cells indicate unavailable data.

these smaller sets, 5 and 7 tRNA genes, respectively, reached study-wide significance. In these sets 5/5 and 6/7 of these were present in 16 study-wide significant tRNA genes.

Furthermore, we examined a subset of samples with longitudinal data ($n = 658$ methylomes from 329 individuals, median age difference 7.6 years). At the nominal significance threshold ($p < 0.05$) this yielded a split of 41 hypermethylating tRNA genes and 22 hypomethylating tRNA genes. Of these hypermethylated tRNAs, 2 are in the previously identified genome-wide significant set of 3 (with tRNA-iMet-CAT-1-4 ranked 3rd by $p$ value) and 9 are in the study-wide significant set of 16. We also ran a number of additional analyses to investigate this directional observation further, including performed linear mixed modelling with batch, lymphocyte, monocyte, neutrophil and eosinophil cell-count data, as fixed effects, and family id and zygosity as random effects. Thus we have observed the same consistent hypermethylation trend with age across a wide array of models, with and without correction for cell-type composition, and when either correcting for structure in our sample population or when examining those sub-populations separately. Full results for each age model are provided in Supplementary Data 1

**tRNA genes are enriched for age-related DNA hypermethylation.** Whilst ageing changes are pervasive throughout the DNA methylome, we identified a strong enrichment for this to occur within the tRNA genes (Fisher's exact test $p = 1.05 \times 10^{-27}$) (see Fig. 2b). This is still significant if the 6 of the study-wide significant 16 tRNAs that have any overlap with polycomb or bivalent regions are excluded ($p = 4.66 \times 10^{-15}$)

CpG density itself is known to have a clear impact on the potential for variability of the DNA methylome as well as ageing-related changes[48,49]. To assess whether this hypermethylation finding was being merely driven by the inherent CpG density of the tRNA genes, we performed a CpG density matched permutation analysis (1000×). This supported the specific nature of these age-related DNAm changes within the functional tRNA genes (Empirical $p$ value $< 1.0 \times 10^{-3}$, Fig. 2a). As a point of comparison for this genomic functional unit, we also performed the same permutations for the known age-related changes in the promoters of genes that are polycomb group targets[8] and those with a bivalent chromatin state[9]. We were able to reproduce the enrichment of the polycomb group targets and bivalent regions (Empirical $p$ value $< 1.0 \times 10^{-3}$) in our dataset.

tRNA-iMet-CAT-1-4 (Fig. 3a) is located in the largest tRNA gene cluster in the human genome at chr6p22.2-1. This cluster contains 157 tRNA genes spanning the 2.67 Mb from tRNA-iMet-CAT-1-2 to tRNA-Leu-AAG-3-1, and also hosts a histone gene microcluster. tRNA-Ile-AAT-4-1 and tRNA-Ser-AGA-2-6 are neighbours and are located on chromosome 17 (Fig. 3b). Notably tRNA-Ile-AAT-4-1 and tRNA-Ser-AGA-2-6 have a third close neighbour tRNA-Thr-AGT-1-2 which does not show significant age-related hypermethylation. We observe a similar pattern of sharp peaks of significance closely localised around the other loci in the study-wide significant set. GENCODE 19 places tRNA-Ile-AAT-4-1 in the 3′UTR of a Nonsense-mediated decay transcript of *CTC1* (CST Telomere Replication Complex Component 1, CTC1-201, ENST00000449476.7) and not of its primary transcript.

To place these hypermethylating tRNA genes in their genomic context we examined how the extended set of 44 non-blood cell-type corrected study-wide significant tRNAs clustered with other tRNA genes. We clustered the tRNA genes by grouping together

## a    Significant Age Related Hypermethylation

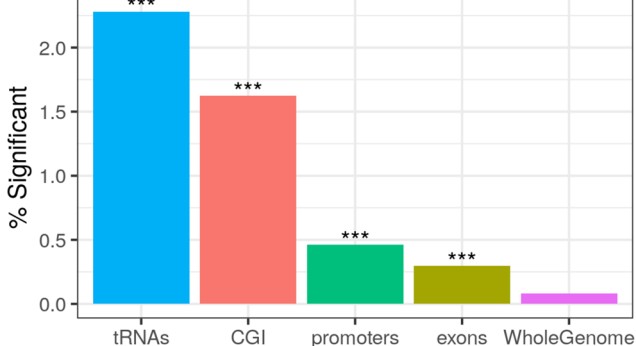

Percentage of significant windows by feature type

## b    CpG Density Permutation Analysis

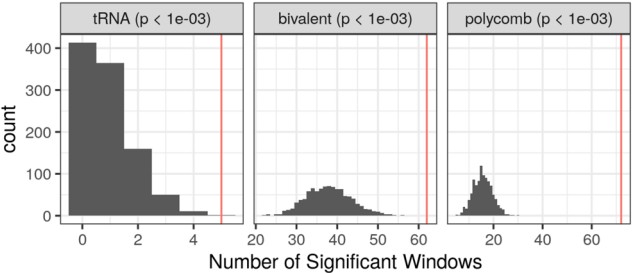

**Fig. 2 Age-related hypermethylation is enriched in tRNA genes relative to genomic background and accounting for CpG density. a** tRNA genes are enriched for age-related hypermethylation compared to the genomic background, Fisher's exact test ($p < 1.05 \times 10^{-27}$, one-sided). All Fisher's exact test results are included in Supplementary data 2 (CGI: CpG islands). **b** tRNA genes show more significant hypermethylated loci than CpG density-matched permutations. Each permutation represented a random set of windows matching the CpG density of the functional unit (tRNA gene loci, bivalent domains, and polycomb group target promoters). These are subsequently assessed for significant age-related DNAm changes, quoted $p$ values are empirical $p$ values based on the permutation results. The red line is the observed number of significant loci. Source data are provided as a Source Data file.

all tRNAs within 5 Mb of one another and then required that a cluster contain at least 5 tRNA genes with a density of at least 5 tRNA genes per Mb. This yielded 12 major tRNA gene clusters containing a total of 353 tRNA genes, with 42 of the 44 study-wide significant tRNA genes within these clusters (Fig. 4a). The hypermethylating tRNA genes are spread evenly among these clusters proportionately to their size (% ageing tRNA per cluster; non-significant one-way ANOVA).

*Age-related tRNA gene set DNA hypermethylation is even observed in one newborn versus one centenarian.* We examined an available Whole-Genome Bisulphite-sequencing (WGBS) dataset from Heyn et al.[50]. These data consisted of blood-derived DNA WGBS in one newborn child and one 26 year old, and centenarian (103 years). In their analysis, the centenarian was found to have more hypomethylated CpGs than the neonate across all genomic compartments, including promoters, exonic, intronic and intergenic regions. However, even in this examination of three individuals of three different ages in the 55% of tRNA genes that possessed coverage, we observed DNA hypermethylation with age among the study-wide significant hypermethylating tRNA genes. The centenarian was significantly more methylated in this set of tRNAs

than the neonate (Wilcoxon rank sum test, 6.14% increase (95% CI -Inf - 4.31), $W = 717$, $p = 6.14 \times 10^{-4}$, see Fig. 4b).

*Age-related changes independently replicated with targeted BiS sequencing.* In order to further robustly support these ageing-related changes, we attempted to replicate these findings ourselves in an independent ageing dataset. Furthermore, we employed a different technology targeted BiS sequencing to also further validate the MeDIP-seq-derived results. These data provide individual CpG resolution to identify what may be driving the regional DNAm changes observed, and precise quantitation of the magnitude of change.

We performed this targeted BiS-seq in blood-derived DNA from 8 pools of age-matched individuals at 4 time points (~4, ~28, ~63, ~78 years) from a total of 190 individuals, as detailed in Table 1. This approach permitted us to assay tRNA gene DNA methylation across a large number of individuals whilst requiring a minimal amount of DNA from each (80–100 ng), and costing ~1/24th as much as performing sequencing individually. A total of 79 tRNA loci generated reliable results post-QC . These tRNAs covered a total of 458 CpGs with a median of 6 CpGs per tRNA (range 1–9). Median coverage per site across pools, technical replicates and batches was 679 reads (mean 5902).

First, we ran the eight Pooled samples on the Illumina EPIC (850k) array to confirm that our pooling approach was applicable for DNAm ageing-related evaluation (Available at GSE166503). This showed an $R^2 = 0.98$ between pool mean chronological age and Horvath clock DNAm predicted age[11] (see Fig. 5c). Therefore, this confirmed the utility of our pooling approach. We also used these array derived data to estimate the major blood cell proportions for each of these pools with the Houseman algorithm[51].

We noted that individual tRNA loci exhibiting age-related changes in DNAm had duplicate or isodecoder (same anticodon but body sequence variation) sequences in the genome, which despite exact or near sequence identity did not show similar changes. tRNA-iMet-CAT-1-4 for instance is 1 of 8 identical copies in the genome and was the only locus that showed significant changes. The results of pairwise differential methylation tests between age groups for the 6 top tRNAs from the MeDIP-seq models are listed in Table 2. tRNA-iMet-CAT-1-4 shows a pairwise increase of 3.7% from age 4 years to age 78 years.

Of the 3 top hits in MeDIP-seq, tRNA-iMet-CAT-1-4 (Fig. 5a (iii)) and tRNA-Ser-AGA-2-6 (Fig. 5a(viii)) exceeded nominal significance ($p$ values = $9.35 \times 10^{-4}$ and $4.28 \times 10^{-2}$, respectively). However, tRNA-Ile-AAT-4-1 (Fig. 5a(xi)) showed a nominal decrease in DNAm with age. tRNA tRNA-Leu-TAG-2-1 from the study-wide significant set also showed nominally significant hypermethylation with age (Fig. 5a(xviii)). Also, four of the individual CpGs in tRNA-iMet-CAT-1-4 exhibited nominally significant increases in DNAm with Age (Fig. 5b).

*Select duplicates and isodecoders of hypermethylating tRNA loci remain unchanged.* We targeted a selection of these duplicate and isodecoder loci for BiS sequencing in order to confirm that the identified DNAm changes are specific to a given locus and not general to related tRNAs. Examining the tRNA-iMet-CAT-1 family, only the previously identified 1-4 version confirmed significant hypermethylation (not 1-2, 1-3 or 1-5) (Fig. 5a(i–iv)). Likewise the tRNA-Ser-AGA-2-6 version was supported compared to 2-1,2-4 and 2-5 (Fig. 5a(v–viii)). Full age model results are available in Supplementary Data 8.

*DNA methylation 450k array data validates the MeDIP-seq results.* Although DNA methylation array poorly covers the tRNA

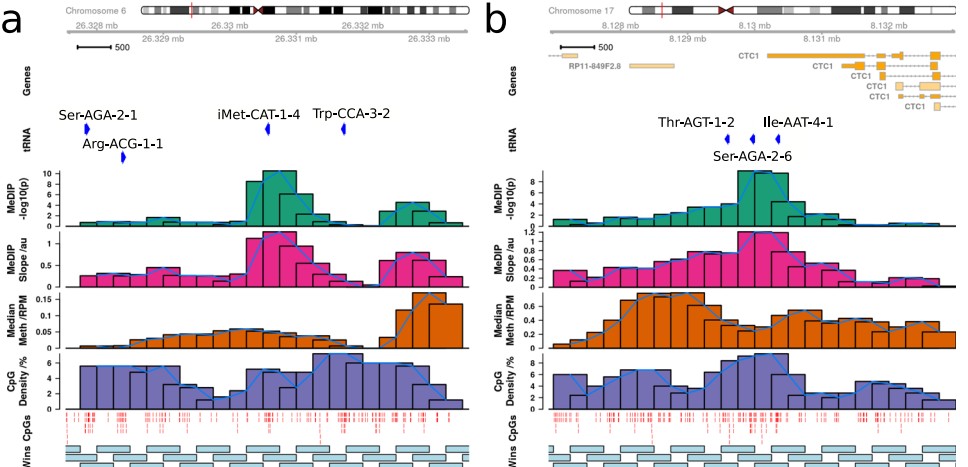

**Fig. 3 Age-related tRNA gene DNA hypermethylation is localised to individual tRNA genes.** MeDIP-seq results (blood cell-type corrected model) for **a** tRNA-iMet-CAT-1-4 as well as **b** tRNA-Ser-AGA-2-6 and tRNA-Ile-AAT-4-1 exhibiting ageing-related DNA hypermethylation. Top to Bottom: chromosome ideogram; gene locations (GENCODE 19); MeDIP-seq DNA methylation ageing $-\log_{10} p$ value results (shown for each 500 bp overlapping window); MeDIP-seq slope of change with age; MeDIP-seq Medium coverage (Reads per Millions, RPM) calculated across all samples; CpG density (%); CpG locations (red lines); 500 bp overlapping windows. The sharp peaks suggest that the results are localised to individual tRNA genes, not the entire genic locus. One window overlapping tRNA-Ile-AAT-4-1 also partially overlaps tRNA-Ser-AGA-2-6. The 3′UTR of the *CTC1* transcript CTC1-201 (ENST00000449476.7, GENCODE 19), which is subject to nonsense-mediated decay, overlaps tRNA-Ile-AAT-4-1. tRNA genes with similar CpG density are exhibiting differing age-related DNAm change patterns. Source data are provided as a Source Data file.

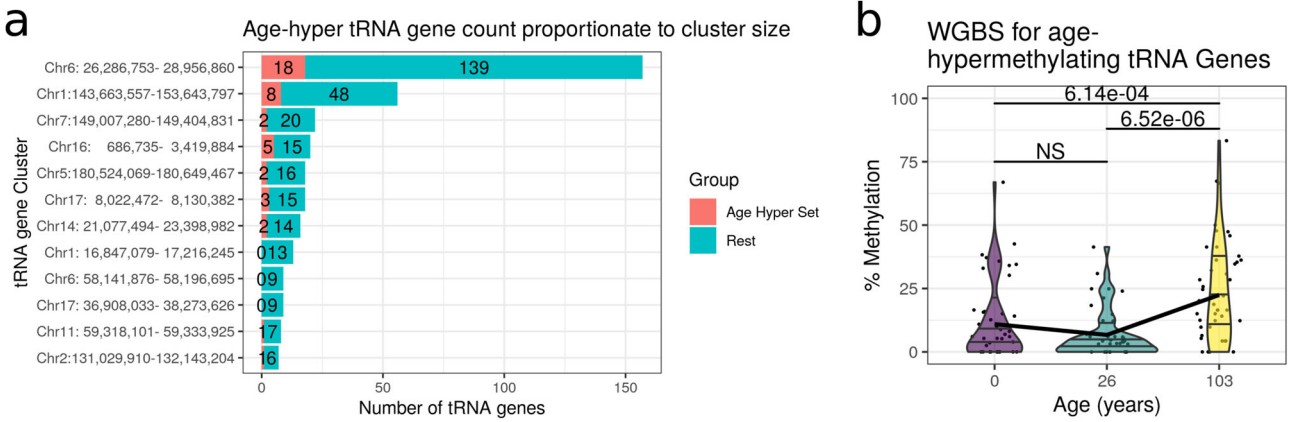

**Fig. 4 Age-hypermethylating tRNAs can be seen when comparing a neonate to a centenarian and are distributed proportionately across major tRNA gene clusters. a** No significant clusters of ageing-related tRNA. Proportion of non-blood cell corrected ageing-related study-wide significant tRNAs (42/44, red) within tRNA clusters (tRNAs within 5Mb with>5 tRNA genes/Mb). **b** Covered study-wide significant tRNA genes ($n = 14/16$) are more methylated in a centenarian than in a neonate or an adult (26 years) in whole-genome bisulfite sequencing (WGBS) data from Heyn et al.[50]. Each point represents individual CpG methylation levels within a tRNA gene. The horizontal lines in the violin plots represent the 25%, 50% and 75% quantiles. NS not significant. Source data are provided as a Source Data file.

| Table 1 Summary information on participants in each pool. Total of 190 participants in 8 pools, at 4 time points: 4, 28, 63, & 78 years. | | | | | |
|---|---|---|---|---|---|
| **Pool** | **Mean age** | **Sex** | **Min age** | **Max age** | **n** |
| Pool 1 | 4.07 | Male | 3.99 | 4.38 | 20 |
| Pool 2 | 4.09 | Female | 3.99 | 4.36 | 20 |
| Pool 3 | 28.07 | Female | 25.87 | 29.80 | 25 |
| Pool 4 | 28.23 | Female | 26.05 | 30.01 | 25 |
| Pool 5 | 63.40 | Female | 62.80 | 63.80 | 25 |
| Pool 6 | 63.26 | Female | 62.70 | 63.70 | 25 |
| Pool 7 | 77.96 | Female | 75.50 | 80.50 | 25 |
| Pool 8 | 77.22 | Female | 74.40 | 80.10 | 25 |

gene set, we wished to attempt to see if this BiS-based but differing and well-established technology was supportive at all of our DNA hypermethylation findings. TwinsUK had available 450k array on 587 individuals, and this platform includes 143 probes, covering 103 tRNAs. All the three top tRNAs in the MeDIP-seq results were covered by this dataset, and 7 of the 16 study-wide significant set. Nine tRNAs show significant ($p < 4.58 \times 10^{-4}$) increases in DNA methylation with age in models corrected for blood cell counts including all three of the three tRNAs identified in the MeDIP-seq as genome-wide significant and 5 of the 7 study-wide significant set present on the array (Fig. 6). Although it should be noted that 56 of these 143 probes are within the non-robust set of Zhou et al.[42], including 1 of the genome-wide, and 1 of the study-wide results (covering tRNA-Ile-AAT-4-1 and

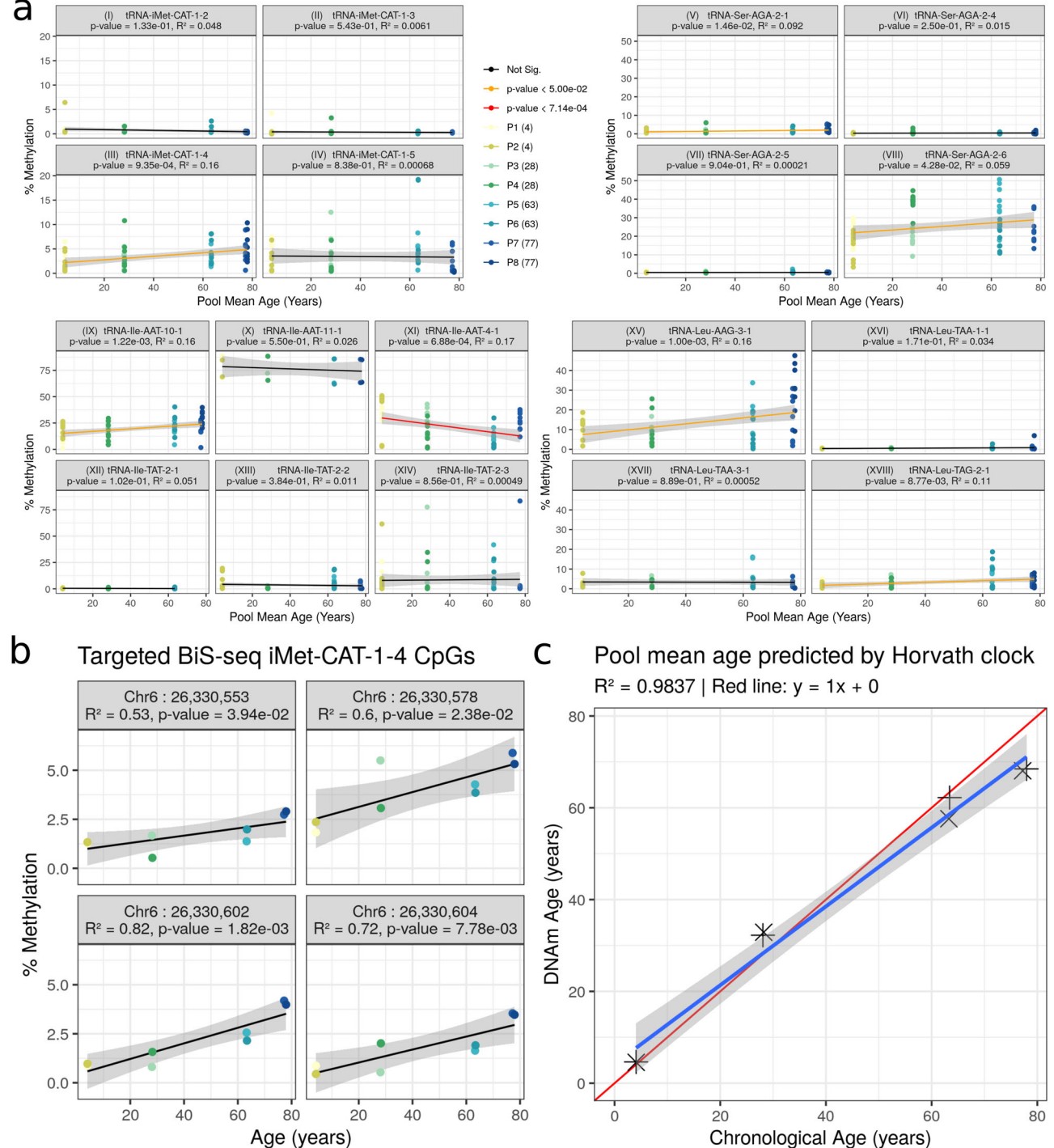

**Fig. 5 Comparison of age-related hypermethylating tRNA genes to closely related tRNA genes using targeted bisulfite sequencing (BiS-seq).**
**a** Combined CpGs within tRNA loci results (experiment-wide Bonferroni $p = 7.14 \times 10^{-4}$); i-iv, Comparison of select tRNA-iMet-CAT loci:
Hypermethylation is specific to iMet-CAT-1-4 (iii) not other isodecoders (i, ii, & iv); v-viii, Comparison of select tRNA-Ser-AGA loci: hypermethylation is specific to Ser-AGA-2-6 A (viii) and to a lesser extent Ser-AGA-2-1 (v), whilst not other isodecoders (vi, vii); ix-xiv, Comparison of select tRNA-Ile loci: Hypermethylation is specific to Ile-AAT-10-1 (ix), Ile-AAT-4-1 (xi) displays hypomethylation contrary to previous MeDIP findings, Ile-TAT-2-2 & 2-3 lack hypermethylation (previously non-significant in blood-corrected MeDIP, although significant in uncorrected), whilst no change in Ile-AAT-11-1 (x) and Ile-TAT-2-1 (xii); xv-xviii, Comparison of select tRNA-Leu loci: Hypermethylation in Leu-AAG-3-1 (xv) consistent with 450k and Leu-TAG-2-1 (xviii) consistent with MeDIP, whilst no change in Leu-TAA-1-1 (xvi) & Leu-TAA-3-1 (xvii). **b** tRNA-iMet-CAT-1-4 Individual CpG analysis: 4 CpGs within this tRNA show consistent methylation increases (all $p < 0.05$). **c** Mean chronological age is tightly correlated with DNAm Horvath clock[11] predicted age for the 8 pooled samples (see Table 1 for pool details). All $p$ values are for F-tests from simple linear regression, Error bands represent the SEM. Source data for these plots are provided as a Source Data file and results for all tRNAs covered are provided in supplementary data 3.

**Table 2 Pairwise differences in methylation between age groups by tRNA p values are for pairwise methylation differences using a paired student's t-test and combined by tRNA region using a generalisation of Fisher's method as implemented the in `RnBeads` v2.0.1 R package[116].**

| tRNA | num. CpGs | comparison | p value | Delta |
|---|---|---|---|---|
| Ile-AAT-4-1 | 8 | 4 vs. 28 | 1.518e−01 | −0.2 |
| | | 4 vs. 63 | 1.774e−01 | −0.234 |
| | | 4 vs. 78 | 3.060e−01 | 0.0113 |
| | | 28 vs. 63 | 7.152e−01 | −0.0334 |
| | | 28 vs. 78 | 1.553e−01 | 0.212 |
| | | 63 vs. 78 | 2.057e−01 | 0.245 |
| iMet-CAT-1-4 | 5 | 4 vs. 28 | 8.403e−02 | 0.0116 |
| | | 4 vs. 63 | 1.716e−01 | 0.0125 |
| | | 4 vs. 78 | 1.997e−04* | 0.0368 |
| | | 28 vs. 63 | 3.943e−01 | 0.000869 |
| | | 28 vs. 78 | 1.724e−02* | 0.0252 |
| | | 63 vs. 78 | 6.224e−02 | 0.0243 |
| Ser-AGA-2-6 | 9 | 4 vs. 28 | 4.222e−01 | 0.0573 |
| | | 4 vs. 63 | 3.968e−01 | 0.0274 |
| | | 4 vs. 78 | 4.651e−01 | 0.0423 |
| | | 28 vs. 63 | 1.095e−01 | −0.0299 |
| | | 28 vs. 78 | 2.126e−01 | −0.015 |
| | | 63 vs. 78 | 2.201e−01 | 0.0149 |

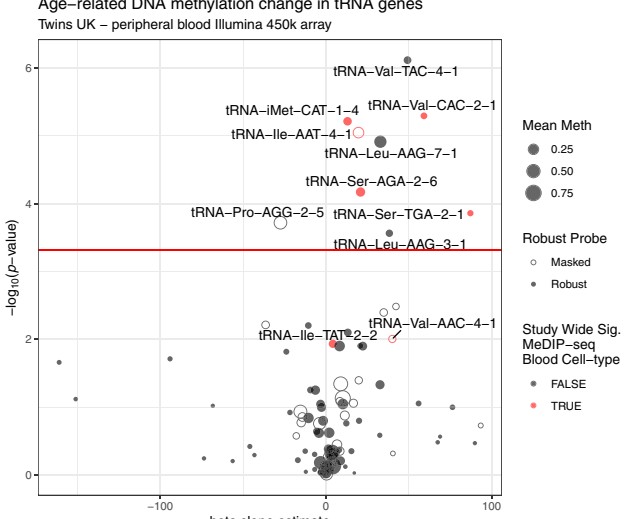

Age–related DNA methylation change in tRNA genes
Twins UK – peripheral blood Illumina 450k array

**Fig. 6 Age-related DNA methylation change in tRNA genes A.** Five of seven tRNAs study-wide significant in MeDIP covered by the 450k array also show significant hypermethylation in these data. tRNAs are labelled if they are significant here or were in the MeDIP-seq data (Red). Model slope: the model coefficient for the methylation values. Unfilled circles indicate those probes with potential issues generated by Zhou et al.[42] this includes tRNA-Ile-AAT-4-1. Significance threshold: $0.05/103 \approx 4.58 \times 10^{-4}$ (the number of tRNA genes examined). Source data are provided as a Source Data file. beta = proportion methylated.

tRNA-Val-AAC-4-1), respectively. Full age model results for all tRNA probes are provided in Supplementary Data 4.

*Ageing-related tRNA loci show increased enhancer-related chromatin signatures.* We further explored the activity of the tRNA genes in public chromatin segmentation data in blood (Epilogos Blood and T cells set)[52,53]. This shows proportionally more

enhancer-related (Enh, EnhBiv & EnhG) chromatin states at tRNA genes hypermethylating with age than the stronger promoter-related (TSS) in other tRNAs. (Fig. 7b). Whereas these characteristics are less frequently predominant in the rest of the tRNAs (Fig. 7b). Age-hypermethylating tRNA are enriched for enhancer chromatin states compared to the rest of the tRNA genes (Fisher's exact test $p = 0.01$).

*Age-hypermethylating tRNAs are more methylated in lymphoid than myeloid cells.* Three tRNA genes remained genome-wide significant and 16 study-wide significant following correction for a major cell-type fraction. This is suggestive of either cell-type independent change or, presumably less likely, a very large effect in a minor cell-type fraction. tRNAs have exhibited tissue-specific expression[54–56] and blood cell-type populations change with age. Specifically, there is a shift to favour the production of cells in the myeloid lineage[47]. These points lead us to examine tRNA gene DNAm in sorted cell populations. We used a publically available 450k array dataset[57] that has been used in the construction of cell-type-specific DNAm references for cell-type fraction prediction using the Houseman algorithm[51]. This consists of data from six individuals (aged $38 \pm 13.6$/years) from seven isolated cell populations (CD4+ T cells, CD8+ T cells, CD56+ NK cells, CD19+ B cells, CD14+ monocytes, neutrophils, and eosinophils). We found that tRNA gene DNAm could separate myeloid from lymphoid lineages (Fig. 7 a and Supplementary Fig. 4).

Of the eight study-wide significant tRNAs with array coverage, we identified that collectively these eight are significantly more methylated in the lymphoid than the myeloid lineage (1.1% difference, Wilcoxon rank-sum test $p = 1.50 \times 10^{-6}$ 95% CI 0.7%–∞). Thus, any age-related increases in myeloid cell proportion would be expected to dampen rather than exaggerate the age-related hypermethylation signal that we observed. In addition, tRNA-Ile-AAT-4-1 and tRNA-Ser-AGA-2-6 have the highest variance in their DNAm of all 129 tRNAs covered in this dataset. This could represent ageing-related changes as these samples range across almost 3 decades. Another possibility may be that these loci as well as hypermethylating are also increasing their variability with age in a similar fashion to those identified by Slieker et al.[58]. In that study, they identified that those loci accruing methylomic variability were associated with fundamental ageing mechanisms.

**tRNA genes also hypermethylate with age in solid tissue.** Some tRNA gene expression has been shown to be highly tissue specific[54–56]. It follows that our observations of changes in DNAm with age in blood might be specific to that tissue. To further investigate this we used a mix of 450k and 27k array data from 'solid tissue normal' samples made available by TCGA (The Cancer Genome Atlas) and data from foetal tissue[59,60] downloaded from GEO. The samples from TCGA range in age from 15–90 ($n = 733$). Only 43 tRNA genes had adequate data to compare across tissues in this dataset and 115 in the foetal tissue data.

Only 2 of the 3 tRNA genes we identified as genome-wide significant and a further 1 of the study-wide significant tRNA genes are present in the set of 45 tRNA genes in the TCGA data, thus limiting our ability to draw conclusions about the tissue specificity of our results. Solid tissue samples have a strong preponderance for low levels of methylation consistent with the active transcription of many tRNA genes and show slight increasing methylation with age but age accounts for very little of the variance (linear regression slope estimate = 1.52; $R^2 = 0.0002$; $p$ value $1.34 \times 10^{-3}$ Supplementary Fig. 5d). In a pan-tissue analysis we found that 10 tRNA genes showed changes

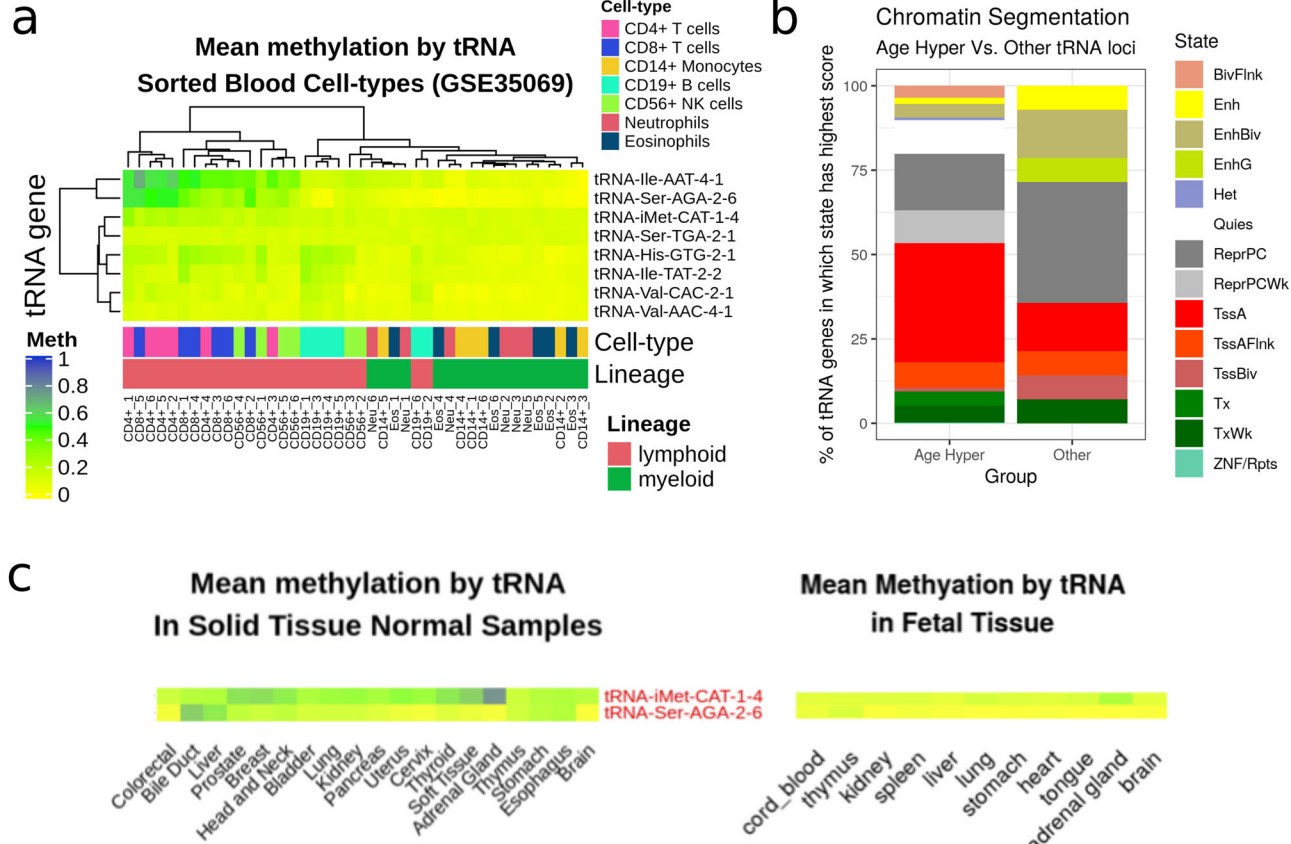

**Fig. 7 tRNA gene DNA methylation in different blood cell types. a** Heatmap of mean DNA methylation of tRNA sorted Blood Cell Types (data from Reinus et al., probes covering 8 of the 16 study-wide significant tRNA in 7 cell-type fractions from 6 males via GSE35069,[57,118]). This illustrates the ability of tRNA gene methylation to separate the myeloid from the lymphoid lineage and the higher methylation in the lymphoid lineage of the 3 cell-type corrected genome-wide significant tRNAs (iMet-CAT-1-4, etc.) **b** Chromatin segmentation data for 'Blood & T-cell' from Epilogos 15 State model[52,53]. Proportion represents the frequency of the predominant chromatin state at a given tRNA (±200 bp) for 14/16 study-wide significant tRNAs covered compared to the other 371 available tRNAs. The ageing-related hypermethylated tRNAs are enriched for enhancer chromatin state (Fisher's exact $p = 0.01$). **c** DNA methylation in 19 tissues from TCGA normal tissue samples and 11 foetal tissues for iMet-CAT-1-4 and Ser-AGA-2-6 (Colour scale as in panel (**a**)). Source data are provided as a Source Data file.

in DNAm with age, 9 of which were hypermethylation ($p$ value < $1.1 \times 10^{-3}$). One of these tRNA genes, tRNA-Ser-TGA-2-1 was also present in a study-wide significant set of tRNA genes. Supplementary Figs. 6 and 7 illustrate minimal tissue-specific differences. Interestingly, however, tRNA-iMet-CAT-1-4 and tRNA-Ser-AGA-2-6 appeared more variable in methylation state than many other tRNAs in the TCGA normal tissue samples (Supplementary Fig. 6) and indeed have the highest variance in DNA methylation across tissues (Supplementary Fig. 5c). These two tRNAs do show broad age-related hypermethylation across a range of tissues in a comparison between foetal to adult, with interestingly, directionally consistent but higher levels for tRNA-iMet-CAT-1-4 in the adrenal gland (Fig. 7c).

**Mice also show age-related tRNA gene DNA hypermethylation.** We examined the DNA methylation of the mouse tRNA genes using data from a reduced representation bisulfite sequencing (RRBS) experiment performed by Petkovich et al.[61]. These data from 152 mice covered 51 tRNA genes and 385 CpGs after QC, representing ~11% of mouse tRNA genes. The mice ranged in age from 0.67–35 months.

Three of the 51 tRNAs showed Bonferroni significant DNA methylation changes with age ($p$ value < $1.08 \times 10^{-4}$) and all were

in the hypermethylation direction. These three are tRNA-Asp-GTC-1-12, tRNA-Ile-AAT-1-4, tRNA-Glu-TTC-1-3 (Fig. 8). Full age model results are available in Supplementary Data 5.

In order to investigate the mouse results further, we made use of data from Thornlow et al.[62], which had established tRNA ortholog sets for 29 mammalian species. They identified 197 mouse tRNAs as having direct human orthologs with 44 of these included in the mouse results from Petkovich et al.[61]. However, unfortunately, although 2 of the top 3 tRNAs (tRNA-Ser-AGA-2-6 & tRNA-Ile-AAT-4-1) have mouse orthologs (tRNA-Ser-AGA-2-2 & tRNA-Ile-AAT-1-1), they are not covered in these mouse data. Furthermore, none of the tRNAs showing significant hypermethylation in mice (at nominal $p < 0.05$) had human orthologs in our MeDIP-seq study-wide significant hypermethylating set. Therefore, whilst we cannot directly compare specific tRNA loci due to this lack of coverage, it is interesting that the small number of significant tRNA genes in the mouse data also hypermethylate with age.

## Discussion

Our work has identified a previously unknown enrichment for age-related epigenetic changes within the tRNA genes of the human genome. This observation was strongly directional with increasing DNA methylation with age[63]. The MeDIP-seq dataset

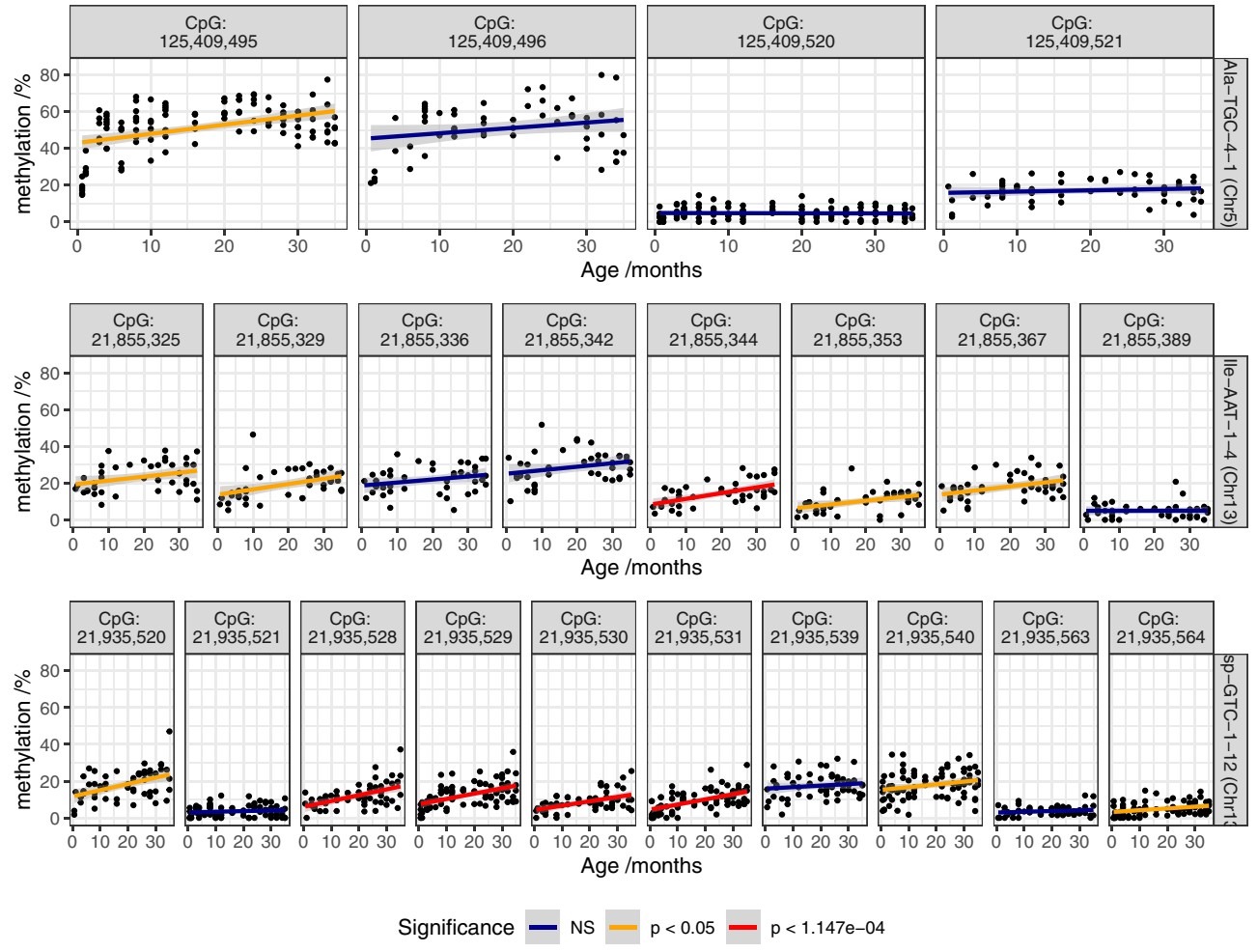

**Fig. 8 DNA methylation of CpGs in 3 tRNA genes which significantly hypermethylate with age in mice.** Data from Petkovich et al.[61]. Six CpGs reach Bonferroni significance and seven show nominally significant increases. Source data are provided as a Source Data file.

employed brought advantages in exploring this undefined terrain of the human tRNA genes. First, being genome-wide it provides much increased access, as these regions are poorly covered by current arrays. Second, being a fragment-based regional assessment of DNA methylation, the individual but highly similar small tRNA genes can be surrounded by unique sequence.

We determined by genome-wide permutation that this strong hypermethylation signal was specific to the tRNA genes, and not merely driven by the underlying CpG density of these loci. A targeted BiS-seq experiment validated the defined nature of the tRNA change in an independent dataset, with a successful pooling approach, which may also be useful for other ageing-related targeted DNA methylome evaluations. In addition, we gained support for our results from limited DNA methylation array data.

Whilst the changes in DNA methylation we have observed are relatively small (i.e., 3.7% between 4-year to 78-year pools in tRNA-iMet-CAT-1-4), this is consistent with the effect size seen in the majority of positions differentially methylatyed with age in many other studies[58,64], except for the noted extreme outliers, such as *EVOLV2*[65]. Furthermore, effect size cut-offs are observed to remove a large fraction of true age-DMPs[66].

We subsequently explored further what was driving this age-related phenomenon and its possible biological implications. As this result was observed in peripheral blood, we were well aware that we were examining DNA derived from a heterogeneous cell-type population[67]. Moreover, that there are well-known age-

related proportional changes in peripheral blood cell composition[47]. The TwinsUK MeDIP-seq and 450k array DNA methylation data included measured haematological values. Therefore, we adjusted for major cell-type effects, such as a myeloid skew, and distinct tRNAs were still significant. Although, a caveat to our study is that this cannot exclude changes in minor specific sub-cell fractions types. However, that these age-related effects were strong enough to be observed in both a regional MeDIP-seq assessment and a pooled sequencing approach, implies that they not extremely subtle. We examined age-related tRNA gene DNA methylation changes in the limited subset of mouse tRNA genes covered in publicly available RRBS data (~13%) and were able to identify tRNAs exhibiting DNA hyper-methylation with age in this set. This suggests that age-related tRNA gene hypermethylation may not be unique to humans, but at least observed across mammals.

Due to the high number of hypermethylating tRNAs prior to cell-type correction, we were also curious whether the epigenetic state of this small tRNA gene fraction of genome could capture and in fact be a defined fingerprint of cell type. We found that tRNA gene DNA methylation could separate myeloid from lymphoid lineages. There was also some suggestion of more fine-grained blood cell-type signatures in tRNA DNAm, such as the separation of CD19+ B cells from CD4/8+ T cells. Ageing is also known to lead to an increase in senescent cells (e.g., CD8+ CD28− cells). Whether these epigenetic changes in the tRNA genes uniquely represent these cell types will

require technical advances to enable future single-cell DNA methylome analysis to accurately assess these regions. If further supported, the epigenetic state of these tRNA loci may aid the taxonomy of cell-type definition.

We observed a predominantly unmethylated state across foetal (Supplementary Fig. 7) and adult tissues (Supplementary Fig. 6) at tRNA gene loci, consistent with the high rate of transcription at many tRNA genes. We suspect that the tRNA genes largely remain unmethylated through development and that the moderate increases in DNAm that we are observing with age at these loci are being driven by changes arising primarily in older individuals. Distinct biological changes have been observed recently in aged individuals[68,69]. Our data are supportive of the tRNA gene set representing a distinct functional unit, which lacks the logarithm change in the young observed strongly in CpGs employed in 'clocks'[13]. This would also be consistent with the lack of significant differences in the tRNA loci detected between the neonate and the 26-year-old adult in the Heyn et al. data. This low baseline DNA methylation of the tRNA genes may also explain why we predominantly observe hypermethylation with age. Whether this is driven by mechanisms, such as aberrant DNA methylation targeting of the tRNA loci or specific sub-cell-type effects with age, will require further experimental investigation.

This signal within the tRNA families was observed to occur at specific Isodecoders. Isodecoders expand in number with organismal complexity and the high prevalence in mammals has been suggested due to their additional regulatory functionality[70,71]. They also have distinct translational efficiency[72], which can also have consequences in human disease[73]. After correcting for major cell types, we identified two tRNA genes tRNA-iMet-CAT-1-4 and tRNA-Ser-AGA-2-6 which had the most consistent hypermethylation across three different assays. Regarding the inconsistent tRNA-Ile-AAT-4-1 result, it is covered by a MeDIP-seq 500bp window which exhibited genome-wide significant hypermethylation, but also partially overlaps tRNA-Ser-AGA-2-6 (Fig. 3B). Whilst the 450k array probe overlapping tRNA-Ile-AAT-4-1 (cg06382303) appears to replicate this hypermethylation, it is a borderline case for exclusion flagged by Zhou et al.[42] due to non-unique 30 bp 3′ subsequence. In the targeted BiS sequencing data, tRNA-Ile-AAT-4-1 exhibited a loss of methylation. These factors considered together suggest that the result for tRNA-Ile-AAT-4-1 should be regarded as inconclusive. Therefore, this may suggest that the hypermethylation signal observed at this locus in the MeDIP-seq data could have been 'pulled up' by the neighbouring tRNA-Ser-AGA-2-6 hypermethylation signal. Of the 16 study-wide significant tRNA genes, only these two of these have a shared significant window, furthermore, in the expanded set of 44 only tRNA-Thr-AGT-1-2 could be similarly affected.

There is great complexity to the fragmentation of tRNA[23], with physiological processes such as stress shown to induce fragment production[74]. These resultant tsRNAs can feedback on protein synthesis by regulating ribosome biogenesis[75] and others have diverse regulatory functions such as targeting transposable element transcripts[76]. They are also observed to circulate in the blood in a cell-free fashion, and fragment levels can be modulated by ageing and calorie restriction[41]. The isodecoder specific nature of our findings frame a possible hypothesis for regulatory change with age and future work will be required to unravel this potential. tsRNA abundance has been associated with locus-specific tRNA gene expression, in some cases independent of mature tRNA levels[77]. This has important implications for the interpretation of our results given the multi-copy nature of genes like tRNA-iMet-CAT-1-4, as even if expression levels of mature iMet tRNAs are unaffected by changes in one copy's DNA methylation, these changes could still influence the levels of particular tsRNAs derived from specific tRNA loci.

The location of tRNA-Ser-AGA-2-6 and tRNA-Ile-AAT-4-1 immediately downstream of CTC1 and of tRNA-Ile-AAT-4-1 within the 3′UTR of an alternate isoform of CTC1, which undergoes nonsense-mediated decay, raises the possibility that the gene body epigenetic regulation of CTC1 may impact on the state of these tRNA genes. CTC1's function in telomere maintenance[78], DNA replication licensing[79], and its role in a rare progeroid condition[80] indicate that it has ageing-relevant biology. The possible relationship between the regulation of CTC1 and that of the tRNA genes downstream of it warrants further study.

Whilst, the expression of the tRNA genes has long been simplified as 'constitutive,' some observations have indicated that many tRNA genes are expressed in a tissue-specific fashion in diverse organisms[55,56]. Although others have found the majority of isodecoders are transcribed in different cell types[28]. Several transcription factors acting via TFIIIB[81] have a negative (the tumour suppressors p53[82] and Rb[83]) or positive (the proto-oncogene c-Myc) influence[81]. Regulatory sequence in the flanking or the internal regions of tRNA genes does not explain tRNA expression variation[84]. Whilst DNAm is able to repress the expression of tRNA genes[32], the broader chromatin environment also affects tRNA transcription. Due to the co-ordinated nature of epigenomic modifications, it may also be revealing to evaluate ageing-related histone modification in these tRNA loci. Furthermore, recent data from Gerber et al.[85] indicates Pol-II may also have a dynamic regulatory role in tRNA expression.

Changes in the epigenetic state of specific tRNA could be modulating transcription efficiency or even codon availability in the ageing cell. tRNA gene dosage is quite closely matched to amino acid usage frequency in the human exome. However, the transcriptome codon usage frequency and tRNA gene expression have been claimed to vary with the replicative state of cells, separating differentiated from replicating cells[86]. Others have argued that these differences are substantially explained by variation in GC content[87] and that codon usage frequencies are observed to be mostly invariant in the transcriptomes of a wide range of tissues, as well as across developmental time[84]. Although, experimental stress-related states have revealed changes with an overrepresentation of codons that are translated by rare tRNAs[88].

tRNA sequences themselves are under strong structural (both secondary and tertiary)[70] as well as functional constraint, which leads to an order of magnitude reduction in variation compared the background genomic mutation rate[28]. Despite strong purifying selection maintaining very low variation in tRNA gene bodies, tRNA genes are subject to high levels of transcription-associated mutagenesis leading to elevated mutation rates over evolutionary time in their immediate flanking sequences[89]. The possible effects of genetic variation on DNA methylation mean that polymorphic tRNA could be another potential caveat to our work. Although, there is no significant population variation in, for example, tRNA iMet sequences in 1000 Genomes data. Indeed, there are only 11 new isodecoder sequences with high confidence (tRNAscan scores ≥50) at >1% population frequency[28]. There is also some evidence for tRNA copy number variation at specific loci, although this remains undercharacterised[90,91]. Another potential cause we considered was whether age-related somatic copy number increases could be occurring in these loci. Population or somatic copy number expansions could lead to increased methylated reads in MeDIP-seq without any epigenetic state change. However, this would not be consistent with the targeted and array BiS conversion methodologies, where the proportion of methylated to unmethylated reads would still be constant.

It is worth noting the parallels with known cancer and ageing epigenetic changes, and that tRNAs are also dysregulated in

cancer[92], with proposed utility as prognostic markers[93]. Furthermore, the early replicating state of tRNA loci, potentially associated with high expression[94], may make them prone to hypermethylate, as is observed in early replicating loci in both cancer[95] and senescent cells[96]. Interestingly, tRNA gene loci may also play a role in local as well as large-scale genome organisation[43,97]. tRNA gene clusters act as insulators[98] and have extensive long-range chromatin interactions with other tRNA gene loci[43]. The co-ordinated transcription of tRNAs at sub-nuclear foci and the B-box sequence elements bound by TFIIIC and not PolIII may represent an organising principle for 3D-chromatin by providing spatial constraints[99]. Therefore, these tRNA epigenetic changes could contribute to the structural changes that are also observed in ageing[100].

In conclusion, due to the unique challenges that make the tRNA genes difficult to examine they have remained epigenetically undercharacterised despite their critical importance for cell function. We directly interrogated the epigenetic DNA methylation state of the functionally important tRNA genes, across the age spectrum in a range of datasets as well as methodologies and identified an enrichment for age-related DNA hypermethylation in the human tRNA genes.

## Methods

**Participants**. Participants in the 'EpiTwins' study are adult volunteers from the TwinsUK Register. The participants were aged between 16 and 82 years, with a median of ~55 years (cohort profile[101]). Ethics for the collection of these data were approved by Guy's & St Thomas' NHS Foundation Trust Ethics Committee (EC04/015—15-Mar-04) and written informed consent was obtained from all participants.

Participants for our targeted BiS sequencing of select tRNA loci were drawn from two studies. Samples from participants aged 4 and 28 years are from the MAVIDOS[102] study and participants aged 63 and 78 years are from the Hertfordshire cohort study[103]. Due to a limited number of available samples, the two 4-year-old pools contained DNA from 20 individuals each, with all other pools having 25 contributing individuals. Pool 1, the first 4-year-old pool used DNA from all male samples, with all other pools using all female samples. Thus, the total number of participants was 190 (see Table 1). Samples from the 28-year-old time point are all from pregnant women at 11 weeks gestation.

**tRNA annotation information**. Genomic coordinates of the tRNA genes were downloaded from GtRNAdb[27]. The two tRNAs located in chr1_gl000192_random are tRNA-Gly-CCC-8-1 & tRNA-Asn-ATT-1-2 (Supplementary Data 6). The 213 probes overlapping tRNA genes were derived from intersecting the tRNA gene annotation data from gtRNAdb with the Illumina 450k array manifest annotation for the hg19 genome build using `bedtools` v2.17.0[104]. We excluded 107 tRNAs from blacklisted regions of hg19[46].

**tRNA gene clustering**. We clustered the tRNA genes by grouping together all tRNAs within 5Mb of one another using the `bedtools merge` tool v2.17.0[104]. We used a command of the form: `bedtools merge -c 4 -o collapse -d N -i hg19-tRNAs.bed` where N is the binsize. We varied the binsize and selected 5Mb as it is at approximately this size that number of clusters with more than one tRNAs exceeds the number of singleton tRNAs (Supplementary Fig. 9). We added the further requirements that these groupings contain at least 5 tRNA genes with a density of at least 5 tRNA genes per Mb to be considered as clusters.

## DNA methylome data

*TwinsUK MeDIP-seq methylomes*. The Methylated DNA Immunoprecipitation sequencing (MeDIP-seq) data were processed as previously described[16,105]. These processed data are available from the European Genome-phenome Archive (EGA) (https://www.ebi.ac.uk/ega) under study number EGAS00001001910 and dataset EGAD00010000983. The dataset used in this work consists of 4350 whole-blood methylomes with age data. Four thousand fifty-four are female and 270 male. Three thousand one have full blood counts. There are 3652 individuals in this dataset. These individuals originate from 1933 unique families. There are 1234 MZ twin pairs (2468 individuals), and 458 DZ twin pairs (916 individuals).

MeDIP-seq used a monoclonal anti-5mC antibody to bind denatured fragmented genomic DNA at methylated CpG sites. The kit used was the 'MagMeDIP' kit (Kit Cat. No.: CO2010021 mc-magme-048 from Diagenode (Liège, Belgium) https://www.diagenode.com/en/p/magmedip-kit-x48-48-rxns), and the monoclonal antibody was antibody 33D3 (C15200081 https://www.diagenode.com/en/p/5-mc-monoclonal-antibody-33d3-premium-100-ug-50-ul). The antibody was

incubated with Adaptor-ligated DNA combining 0.5 µl antibody + 0.5 µl water; then 0.6 µl MagBuffer A, 1.4 µl water and, 2 µl MagBuffer C; yielding a final volume of 5 µl for the immunoprecipitation reaction. Validation information including for the use of this antibody in MeDIP is provided on the manufacturer's website in the datasheet for this antibody (https://www.diagenode.com/files/products/antibodies/Datasheet_5-mC33D3_C15200081-100.pdf) This antibody-bound fraction of DNA was isolated and sequenced[44]. MeDIP-seq 50-bp single-end sequencing reads were aligned to the hg19/GRCh37 assembly of the human genome and duplicates were removed. MEDIPS (v1.0) was used for the MeDIP-seq specific analysis[106]. This produced reads per million base pairs (RPM) values binned into 500 bp windows with a 250 bp slide in the BED format, resulting in ~12.8 million windows on the genome. Thus all individual tRNA genes are covered by 2-3 windows. tRNAs were considered significant if any window overlapping them showed significant changes. MeDIP-seq data from regions of interest were extracted using Bedtools v2.17.0[104].

*Analysis of DNA methylome data for significant ageing-related changes*. All analysis was performed in R/3.5.2. Linear models were fitted to age using the MeDIP-seq DNA methylome data, as quantile normalised RPM scores at each 500 bp window. Models were fitted with: 1. No covariates; 2. Batch information as a fixed effect; 3. Blood cell-type counts for neutrophils, monocytes, eosinophils, and lymphocytes as fixed effects; and 4. Batch and Blood Cell counts as fixed effects. Model 1 & 2 were fitted on the full set of 4350 as batch information was available for all samples but blood cell-count data was only available for a subset of 3001 methylomes. Models 1 & 2 fitted in the $n = 3001$ subset were similar to those fitted in the complete set of 4350. Models 3 & 4 were fitted in the $n = 3001$ subset with full covariate information and sets of significant tRNAs identified at study-wide and genome-wide levels in model 4 were used in subsequent analyses. Models were also fitted for two unrelated subsets created by selecting one twin from each pair (monozygotic (MZ) or dizygotic (DZ)), yeilding sets with $n = 1198$ and 1206 DNA methylomes. An additional model was fitted for longitudinal analysis, samples were selected by identifying individuals with a DNA methylome at more than one time point and filtering for only those with a minimum of 5 years between samples. This yielded 658 methylomes from 329 individuals with age differences of 5–16.1 years, median 7.6 years. Models for this set included participant identifier as a fixed effect in addition to blood cell counts and batch information. The mixed effects model was also fit to include the effect of family structure and zygosity ($n = 2989$). The null model included batch, lymphocyte, monocyte, neutrophil and eosinophil cell-count data, as fixed effects, and family id and zygosity as random effects, and was compared to the same model with the addition of the quantile normalised methylation score using anova, mixed models were fit with `lme4` v1.1.21

*Permutation analysis for enrichment with age-related changes*. We performed a permutation analysis to determine whether the CpG distribution of sets of the tRNA genes was the principle driver of the ageing-related changes observed. Windows overlapping tRNAs have a higher proportion of windows with a greater CpG density than their surrounding sequences (see Supplementary Fig. 8). CpGs residing within moderate CpG density loci are the most dynamic in the genome[48] and CpG dense CpG island regions include specific ageing-related changes[8,9,16]. For comparison we also performed the permutation in the CGI regions from the Polycomb group protein target promoters in Teschendorff et al.[8] and bivalent loci from ENCODE ChromHmm 'Poised Promoter' classification in the GM12878 cell-line[53]. A random set of 500 bp windows representing an equivalent CpG density distribution of the feature set in question were selected from the genome-wide data. The number of windows selected in each permutation for these features was; for tRNAs 1320, for bivalent chromatin 42,841 and for polycomb group protein target promoters 13,502. Above a certain CpG density there are insufficient windows to sample without replacement within a permutation. Furthermore, above ~≥18% CpG density, CpG Islands become consistently hypomethylated[107]. Therefore, all windows with a CpG density of ≥18% (45 CpGs per 500 bp) were grouped and sampled from the same pool. i.e. a window overlapping a tRNA gene which had a 20% density could be represented in permutation by one with any density ≥18%. This permutation was performed 1000 times to determine an Empirical $p$ value by calculating the number of times the permutation result exceeded the observed number of significant windows in the feature set. Empirical $p - value = \frac{r+1}{N+1}$, where $r$ is the sum of significant windows in all permutations and N is number of permutations[108].

*Neonate and centenarian whole-genome BiS sequencing*. DNA methylation calls were downloaded from GEO: GSE31263 and intersected with tRNA genes using bedtools v2.17.0[104].

*Sample pooling and EPIC array*. We performed an Illumina Infinium DNA methylation EPIC array ((C) Illumina) and targeted BiS sequencing of select tRNA gene loci. Here we used DNA extracted from whole blood and pooled into 8 samples from unrelated individuals at 4 time points with 2 pools at each time point. The time points were 4, 28, 63, and 78 years. Using the EPIC array we were able to infer the DNAm age using the Horvath DNAm clock[11] and blood cell-type composition of our samples using the Houseman algorithm[51], as implemented in the `meffil` v1.1.1 R package using the 'blood gse35069' cell-type reference option[109].

*Targeted BiS sequencing*. We selected tRNA loci for targeted sequencing in which we had observed DNAm changes with age and closely related tRNAs in which changes were not observed. Primer design was performed using 'methPrimer' v1[110] (Supplementary Data 7). A total of 84 tRNA loci were targeted and 79 subsequently generated reliable results post-QC. The targeted tRNAs covered a total of 723 CpGs with a median of 8 CpGs per tRNA (range 1–13), data passing QC was generated for 458 CpGs, median 6 (range 1–9) per tRNA.

Quality was assessed before and after read trimming using `fastqc` v0.11.5[111] and `multiqc` v1.0[112] to visualise the results. Targeting primers were trimmed with `cutadapt` v1.13[113] and a custom perl5 script. Quality trimming was performed with `trim_galore` 0.5.0[113,114]. Alignment and methylation calling was performed with `Bismark` v0.20.0[115] making use of `bowtie2` v2.3.1[116]. The alignment was performed against both the whole hg19 genome and just the tRNA genes ±100 bp to assess the possible impact of off-target mapping. Mapping to the whole genome did produce purported methylation calls at a larger number of loci than mapping just to the tRNA genes (683,783 vs 45,861, respectively). Introducing a minimum coverage threshold of 25 reads dramatically reduced this and brought the number of sites into line with that in the tRNA gene set (36,065 vs 33,664 respectively) suggesting a small number of ambiguously mapping reads. All subsequent analysis was performed using the alignment to just the tRNA genes with a minimum coverage of 25 reads.

We performed pairwise differential methylation analysis of the tRNA genes at the different time points using `RnBeads` v2.0.1[117] with `limma` v3.38.3[118] and a minimum coverage of 25 reads. We also performed linear regression predicting age from DNA methylation at the targeted tRNA sites, permitting us to compare rates of increase with age. For the linear regression, we used only CpG sites with more than 25 reads mapped to the regions of the genome targeted for amplification.

*TwinsUK Illumina 450k array methylomes*. Illumina Infinium DNA methylation 450k arrays ((C) Illumina) were also performed on TwinsUK participants, in 770 Blood-derived DNA samples which had matched MeDIP-seq data. These data were pre-processed in the form of methylation 'beta' values pre-processed as previously described[16,105]. Cell-type correction was performed using cell-count data and the following model: `lm(age ~ beta + eosinophils + lymphocytes + monocytes + neutrophils)`.

**Chromatin segmentation data**. Epilogos chromatin segmentation data[52] were downloaded for the tRNA gene regions ±200 bp from https://explore.altius.org/tabix/epilogos/hg19.15.Blood_T-cell.KL.gz using the `tabix` v1.3.2 utility. The data used were the 'Blood & T-cell' 15 State model based on segmentation of 14 cell types. These data were manipulated and visualised with R v3.5.2 and ggplot2 v3.2.1.

**Isolated blood cell-type specific data**. Data from seven cell-type fractions from six Male individuals was downloaded from GSE35069[57] using GEOquery v2.51.5[119]. Five of the 6 top age-hypermethylating tRNAs are covered by this array dataset. Heatmaps were created with the `ComplexHeatmap` v1.20.0 R package[120].

**Cancer and tissue-specific methylation data**. Data were downloaded from the TCGA via the GDC (genomic data commons) data portal[121] using the `GenomicDataCommonsR` package v1.6.0. Data from foetal tissue[59,60] were downloaded from GEO (GSE72867, GSE30654). From the TCGA, we selected samples for which DNAm data were available from both the primary site and normal solid tissue, and for which we could infer an approximate age (within 1 year). We selected those probes overlapping tRNA genes yielding 73,403 data points across 19 tissues with an age range of 15–90 years (median 63.4) (Supplementary Data 8)

**Mouse RRBS analysis**. We downloaded methylation calls and coverage information resulting from RRBS performed by Petkovich et al.[61] from GEO using GEOquery v2.51.5[119] from GSE80672. These data from 152 mice covered 68 tRNA and 436 CpGs after QC requiring >50 reads per CpG and >10 data points per tRNA. We excluded 5 tRNAs from blacklisted regions of mm10[46]. After QC there were 58 tRNA genes and 385 CpGs. We performed simple linear modelling to predict age from methylation level at each tRNA and each CpG.

**Reporting summary**. Further information on research design is available in the Nature Research Reporting Summary linked to this article.

## Data availability

All data used in this study is publicly available under the following accession numbers.

Targeted Bisulfite sequencing data is deposited in the sequence read archive with the bioproject accession: PRJNA635108.

EPIC array data is deposited at GEO with the accession: GSE166503.

Neonate and Centenarian Whole-Genome Bisulfite Sequencing DNA methylation calls: GSE31263.

Human isolated blood cell-type specific DNA methylation array Data: GSE35069.

Cancer and tissue-specific DNA methylation array data from TCGA, see (Supplementary) for the full list of samples drawn from genomic data commons.

Foetal tissue DNA methylation data were downloaded from: GSE72867 and GSE30654.

Mouse whole blood RRBS DNA methylation data data: GSE80672.

The MeDIP-seq data supporting the results of this article are available in the EMBL-EBI EGA under Dataset Accession number EGAD00010000983, access is subject to request and approval by their Data Access Committee.

Epilogos chromatin segmentation data are available from: [https://explore.altius.org/tabix/epilogos/hg19.15.Blood_T-cell.KL.gz]

Twins UK DNA metylation and age model summary data for non-overlapping 500 bp windows is available at via UCSC Genome Browser's track hub interface, add: http://epigenome.soton.ac.uk/TrackHub/hub.txt Tracks include: mean, median and variance in RPM values across all samples in the model ($n = 3001$); the percentage of samples with an exactly 0 RPM score in a given window (useful for spotting technical issues); the slope and $-\log10(p$ values) for batch corrected, and blood cell-type corrected age models.

tRNA gene annotations for the hg19 and mm10 genomes where acquired from GtRNAdb [http://gtrnadb.ucsc.edu/]

Source data are provided with this paper.

## Code availability

Available at https://github.com/RichardJActon/tRNA_paper_code https://doi.org/10.5281/zenodo.4294046[122]

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

## Acknowledgements

We gratefully acknowledge the individuals from TwinsUK, Mavidos and the Hertfordshrie cohort. TwinsUK received funding from the Wellcome Trust (Ref: 081878/Z/06/Z), European Community's Seventh Framework Programme (FP7/2007-2013), the National Institute for Health Research (NIHR)-funded BioResource, Clinical Research Facility and Biomedical Research Centre based at Guy's and St Thomas' NHS Foundation Trust in partnership with King's College London. Further funding support for the EpiTwin project was obtained from the European Research Council (project number 250157) and BGI. SNP Genotyping was performed by The Wellcome Trust Sanger Institute and National Eye Institute via NIH/CIDR. The authors would like to thank Nikki Graham for her assistance with the identification and pooling of the MAVIDOS and Hertfordshire DNA samples. The authors also acknowledge the use of the IRIDIS High Performance Computing Facility, and associated support services at the University of Southampton, in the completion of this work. The MRC-LEU is supported by the Medical Research Council (MRC). R.J.A. was in receipt of a MRC Doctoral fund (1820097).

## Author contributions

R.J.A. designed experiments and analysed all the processed and experimental data. C.G.B. conceived and designed the experiments. T.D.S. and J.W. conceived and provided TwinsUK MeDIP-seq data. Y.X., F.G. and J.W. produced raw MeDIP-seq data with WY and JB processing and quality controlling these data. WY contributed an analysis concept. C.C., N.C.H., E.D., and K.L. provided MAVIDOS and Hertfordshire sample data. E. B., E.W., and C.A.M. performed the targeted BiS sequencing experiment. P.G.H. contributed additional data and discussion of results. R.J.A. and C.G.B. wrote the paper. All authors reviewed and approved the final manuscript.

## Competing interests

C.C. has received lecture fees and honoraria from Amgen, Danone, Eli Lilly, GSK, Kyowa Kirin, Medtronic, Merck, Nestle, Novartis, Pfizer, Roche, Servier, Shire, Takeda and UCB outside of the submitted work. N.C.H. reports personal fees, consultancy, lecture fees and honoraria from Alliance for Better Bone Health, AMGEN, MSD, Eli Lilly, Servier, Shire, UCB, Consilient Healthcare, Kyowa Kirin and Internis Pharma, outside the submitted work. E.D. has received speaker/consultancy fees from Pfizer, UCB and Lilly. T.D.S. is a consultant to Zoe Global Ltd ('Zoe'). All other authors declare no competing interests.
