## [Peer Review File · Nature Communications]

Reviewers' comments:

Reviewer #1 (Remarks to the Author):

Acton and coworkers provide a very detailed and comprehensive analysis of aging-related DNA methylation changes in tRNAs. The manuscript is very well written and combines various complementary datasets (including MeDIP-seq, EPIC array, and targeted bisulphite sequencing). The authors describe highly significant DNA hypermethylation at various sites in the genome. However, the difference in DNA methylation at these sites is always less than 5% between young and old samples. The relevance of these changes is therefore questionable and a clear correlation with tRNA expression could not be provided. Overall, the findings are new and might be of interest to many researchers. The terminology should be adjusted since there is little evidence for "strongly directional hypermethylation with advancing age" (last line of abstract). I have the following specific comments:

1) It has been shown that about 15 – 30% of all CpG sites in the genome are associated with age (they become significant in very large cohorts despite very small absolute changes). The finding of some aging-related DNA methylation changes in tRNAs is therefore not surprising. The very small differences in the absolute level of DNA methylation with age therefore need to be more critically discussed (possibly also in the abstract).

2) In Fig. 6 the authors present WGBS data of CpG methylation across tRNAs in newborn, adult and centenarian. How do the authors explain that there are no significant differences between newborn and adult, albeit aging-associated DNA methylation generally follows a logarithmic trajectory in childhood?

3) In line 147 the authors mention their "novel pooling approach" of a total of 190 blood samples in four age categories. It is not entirely clear what is novel about this approach and the results would be more convincing by independent targeted bisulfite sequencing of individual samples.

4) Apparently several tsRNAs reveal moderate expression changes – yet, with no evident overlap related to DNA methylation. How heterogeneous is the expression level within a given sample (particularly in the small subset of cells where DNA methylation is gained)? Would differential expression of tsRNAs impact on the aging process? These aspects might at least need further discussion.

5) The manuscript comprises 14 figures which are sometimes difficult to read. Some of them might be combined into multi panel figures, better labeled (e.g. TRUE and FALSE is not intuitive), color codes should be explained (e.g. Fig 6), and some aspects might be moved to the supplement to make the figures more reader friendly (e.g. the ideograms in Fig 9).

Reviewer #2 (Remarks to the Author):

The authors claim that some tRNA genes, as has been shown for some protein coding genes, are also subject to changes in methylation state as a function of age. This is a novel result and will be of interest to others in both the tRNA and DNA methylation field. While the content is original and I believe important, it was extremely frustrating to wade through the very draft-like Results section, which had an excessive number of figures (14!) and too few tables with *specific* lists tRNA genes being discussed. Some figures are either not necessary or not well-designed (Figure 5), or just wrong (Figure 2)). The abstract was 3 times the maximum length for this journal. I normally do not comment on article length, but because so much editing will be required to get this into a format that is publishable, it puts the burden on reviewers to sort through it all, and it will require a fair amount of revision before final consideration for publication in this forum. Because of the importance and general scientific value of this story, I will be happy to review a revision. For now, here are the positives and the parts that need more work.

I generally appreciated both the introduction and discussion sections, which hit on important parts

of prior published work and the implications of this work. The Results however haven't been carefully edited, and it seems to be a "throw everything on the wall and see what sticks" draft. For a tRNA researcher who cares about the specific genes being discussed, it was frustrating not having the complete list of tRNAs being discussed, beyond the 2-3 that were found to be consistently significant after cell-type correction. In the beginning of the Results section (line 89) "We identified 21 genome-wide significant and 44 study-wide significant results"... please give a summary table of the 21 and 44 tRNA genes, by name, and the different data sets that do or do not support them. Far too often, P-values were given, and gene counts given, but no actual reference to a complete table with the full set of genes mentioned, which is what at least half your audience will really care about. Simply referring to a summary Figure 5 (in Results, line 96), without specific gene names, is not very helpful.

The authors have been appropriately conservative in their assessment of the significance of DNAm changes. What is surprising, though, is the particular tRNA genes that appear to be most significantly affected. Two of the three most significant genes are located within the 3' UTR of the CTC1 gene, only 300 bp apart, which puts them in the same epigenetic context as a third tRNA gene, Thr-AGT-1-2, only 300 bp away from the other CTC1-overlapping genes. I am concerned that the methylation dynamics of all three of these tRNAs are influenced by the overlapping CTC1 gene -- which does not necessarily cast doubt on the results, but is surprising and not what I would have expected. Why isn't Thr-AGT-1-2 also in the same significance bin as the other two? The third significant tRNA, iMet-CAT-1-4, is only about 1kb away from Trp-CCA-3-2, and appears to be in the same epigenetic neighborhood, but is not found to be significant. Again, this makes me curious why these genes are not getting similar significance as those extremely nearby, and should be addressed or acknowledged by the authors.

The authors don't do a particularly good job of explaining or contextualizing why some loci are being hypermethylated, and others hypomethylated, and what the physiological implications of two very different changes over time might have on the aging phenotype. For example, there was limited discussion or interpretation of the puzzling results of Ile-AAT-4-1 having a negative BiS-seq slope. Again, with Ile-AAT-4-1 is just 300bp from Ser-AGA-2-6, why are the BiS-seq slopes so different? Glossing over this (apparently) contradictory data is very troubling and will draw questions from anyone reading this carefully.

As mentioned earlier, the authors don't focus as much as I would hope / expect on the genomic neighborhoods of the tRNA genes this study has found, which seems like a missed opportunity to interpret why these tRNAs, and not others that are in equally "highly transcribed" regions, are so affected. Two of the three overlap a critically important and interesting CTC1 gene, and there is no further mention of this?(!) By focusing on just the 3 most significant tRNA genes for much of the downstream analyses, perhaps some broader understanding is being missed by not looking at patterns among the broader 21 and 44-tRNA sets. As long as the authors are consistently clearly which tRNAs are significant in which statistical framework, it is fine to mention and discuss the broader sets, given that these might become more significant with broader studies in the future. For example, what is the proximity of the 21/44 tRNAs to protein coding or other tRNA genes? As someone who studies human and other mammalian tRNA genes in great detail, I can't understand why just those tRNA genes rose to significance and not many others that show tissue or cellular response specificity. I do not question the conservative methylation results, but feel as though this is only a very partial picture of what is happening in aging because from a tRNA biology perspective, it doesn't give readers any leads on why these loci are strongly affected and not others.

While the methylation results were intriguing and generally convincing, the attempt to assess transcription, and relate that to gene loci was extremely weak. There is an immense amount of (much uncharacterized) processing which creates tRNA-derived small RNAs from full length tRNAs, and it is becoming clear that full-length tRNA abundance is not reflected in fragment abundance in most cases. While I applaud the heroic attempts to convert MINTbase transcription data to something insightful, it is ultimately not scientifically convincing. Angiogenin and a host of RNases create tRNA fragments in response to cellular conditions, and some tRNA fragments are stabilized (and accumulate) thanks to interactions with other proteins in the cell. Thus, aside from the "many to many" fragment to full-length tRNA to tRNA locus mapping gauntlet, the data presented in this

section are off-topic and probably wrong (you *must* use full-length tRNA sequencing methods like DM-tRNA-seq to have any chance to assess full-length tRNA abundance of specific transcripts). I recommend that this part be dropped from the manuscript because it dilutes the scientific rigor of earlier conclusions. Similarly, trying to use tRNA fragment data to assess pre-tRNA abundance, and to estimate full-length tRNA just doesn't work (we've tried with much better data).

The brief section on mouse tRNA epigenetics seemed half-hearted. Do any of the mouse genes identified correlate with the syntenic or orthologous human genes that are in the 3/21/44 gene sets? A recent publication that has a map of tRNA gene synteny might be helpful (PMID: 31857444). If not, can you better explain or relate your mouse findings to the results with human tRNA genes? Right now, this section seems incomplete and does not meet the same standards as the earlier work.

Other points:

It is entirely up to the authors, but using the term "tRNAome" frequently (while, yes, it is mentioned in other papers) is about as elegant as referring to all protein genes as the "mRNAome". I'd recommend simply "tRNA gene set" or "all tDNA" in its place, but that is just a suggestion.

There are major errors in Figure 2, where the gene count numbers are completely wrong in many cases (e.g., Arg and Asn tRNA counts have flipped to the wrong anticodons — Asn GTT and Ala AGC are the most common tRNA genes, yet they are listed as single copy). Also, the available information about high-confidence tRNA genes was not used, which we see as particularly relevant here. In a manuscript with way too many figures, I'd recommend just dropping it because there are no insights drawn by essentially an (inaccurate) re-publication of GtRNAdb information. Also, gene counts alone are extremely misleading, and scientifically mean almost nothing without an understanding of which loci are active and which are not (see PMID: 31857444).

Figure 5 has way too much information to be absorbed, and is very difficult to get a "summary" impression from — just too much data squished down into too little room. For important comparisons between data sets, I would represent in simplified comparisons and possibly moving this to the supplement unless it clearly illustrates key points in the manuscript (it only seems to be referred to once in the main text).

Results, line 314: Ref #29 is cited, but a much more current and thorough analysis of tRNA gene mutation rate can be found in Thornlow et al., 2018 (PMID: 30127029).

Figure 11: Epilogos are cited, but perhaps the contribution of ChromHMM, from which these states were actually computed, should also be cited here (ref #98 in manuscript).

Abstract, line 10: Please use current numbers available at the GtRNAdb — this number is very outdated.

General comment on all Figure legends: it is helpful to state, either in the title of the figure / table, or in the text, what is the scientific point that is most important about the figure — just labeling "Heatmap of X and Y..." is not very helpful to the reader, but "Heatmap showing three tRNAs are significant across tissues" is helpful in guiding the reader through complex figures of from large data sets. Please carefully select the main text figures that illustrate key points, and put the rest / full data set figures in the supplement.

In summary, there is a solid paper with important findings within the current drafty Results section of this manuscript, but it needs to be carefully edited, and there needs to be more attention to what the results might mean biologically (e.g., why mostly hypermethylation, not more hypomethylation? Why is Ile-AAT-4-1 going in the opposite direction in BiS-seq data? What would be the impact on tRNA abundance and translation, if any, if these are multi-copy genes??). There is surprisingly little "tRNA biology", genome context, or discussion of why these particular genes might be affected (and not other loci), which detracts from the potential impact of this paper. The combination of multiple types of data sets, careful correction for cell type variability, and

examination of duplicate / other isodecoder loci are very much appreciated (Fig 9), but the results are still too fragmentary leaving precious few tRNA genes that are shown to be affected by aging. For a gene set that is so large and important (and highly transcribed), just focussing for the most part on three genes that have no thread among them (aside from two overlapping CTC1, not really discussed either), was disappointing. I encourage the authors to hone down the story to more than a collection of statistically significant loci and it will make for much more compelling, impactful work which I look forward to seeing published in a revised state.

Respectfully,
Todd Lowe
UC Santa Cruz

Responses to Reviewers' Comments

We thank the reviewers for their salient and insightful comments on our manuscript. We address all the points raised in an itemised fashion below, with any resulting changes to the manuscript documented alongside them. We hope that the changes made will reassure the reviewers that their time was well spent as we believe they have resulted in substantial improvements to the manuscript.

Reviewer #1

The authors describe highly significant DNA hypermethylation at various sites in the genome. However, the difference in DNA methylation at these sites is always less than 5% between young and old samples. The relevance of these changes is therefore questionable and a clear correlation with tRNA expression could not be provided. Overall, the findings are new and might be of interest to many researchers. The terminology should be adjusted since there is little evidence for “strongly directional hypermethylation with advancing age” (last line of abstract).

We thank the reviewer for this comment and agree with their observation that the DNA methylation differences described are not large swings in methylation state. However, this is consistent with the vast majority of ageing-related changes that have been observed in human ageing studies across the genome to date (e.g. majority age-DMPs <1.25% change/10 years, Sliker et al.¹; average change age-CpGs <1%/10 years, Xu et al.²). Furthermore, using an effect size threshold has been shown to remove a large fraction of true age differentially methylated positions (age-DMPs)³. The robust changes implemented in the strongly replicated Horvath clock⁴ are of the scale of ~3.2% between young (<35 years) to old (>55 years)⁵. Whilst we have identified small magnitude changes, it should be noted that these have been observed with two fundamentally different methodologies with distinct strengths in assessing regional and single-base pair differences, by immunoprecipitation and bisulphite chemical conversion, respectively⁶. Finally, it is the strong directionality, not the magnitude, of our finding that we wished to stress with our phrasing ‘strongly directional hypermethylation’. All of our significant changes in DNA methylation at tRNA gene loci in our MeDIP-seq data were observed to be an increase in DNA methylation and this directional enrichment was supported in our validation and replication data. However, we concede that this could be misinterpreted as implying a large magnitude of change and have therefore rephrased our abstract by removing ‘strongly directional’.

This study is the first comprehensive evaluation of the genomic DNA methylation state at human tRNA genes, and reveals a discreet ~~and strongly directional~~ hypermethylation with advancing age.

We added the following line to the Results to illustrate the effects size for one of our most robust results:

tRNA-iMet-CAT-1-4 shows a pairwise increase of 3.7% from age 4 years to age 78 years.

- 1) It has been shown that about 15 – 30% of all CpG sites in the genome are associated with age (they become significant in very large cohorts despite very small absolute changes). The finding of some aging-related DNA methylation changes in tRNAs is therefore not surprising. The very small differences in the absolute level of DNA methylation with age therefore need to be more critically discussed (possibly also in the abstract).

We agree with the reviewer that identifying age-related DNA methylation changes is not particularly unusual, as these occur widely across the genome. However, what we did identify as surprising was the distinct enrichment within the tRNA gene set (46kb, ~0.002% of the human genome) and, furthermore, as mentioned above, their strong directional skew towards hypermethylation. We took measures to quantify the likelihood of this observation. We compared the proportions of loci significantly hypermethylating with age in other functional elements (Figure 3B in the paper and see B below). Additionally, this tRNA gene signal was not found to be due to overlap with known age-related hypermethylating regions, the Polycomb target gene promoters or Bivalent domains. Also, we hypothesized that the background CpG density of the tRNA gene set could be contributing to this observation - due to the hypermethylation with age observed in CpG islands⁵ as well as MeDIP-seq methodology having the potential to act more efficiently in CpG-rich regions⁶. Therefore, we performed a permutation analysis, which found that the tRNA genes were enriched for age-related hypermethylation even when compared to CpG density matched regions of the genome (Figure 3A in part and A below).

The reviewer's point about small absolute changes in DNAm is well taken. However, as we discussed above, effect size cut-offs are observed to remove a large fraction of true age-DMPs³. Also, we note that some of the many robust replicated environmentally changes in DNA methylation identified to date are of small effect size (e.g. GPR15 (cg19859270) 2.6% decrease from non-smokers to smokers⁷). We found a pairwise increase in DNA methylation across the CpGs in tRNA-iMet-CAT-1-4 of 3.7% from our 4-year-old to our 78-year-old pool samples in our targeted bisulfite sequencing. In addition, the increase in mean methylation of the blood cell-type corrected study-wide significant tRNA gene set from the 26-year-old to the 103-year-old from the Heyn et al.⁸ whole genome bisulphite sequencing (WGBS) data was 15.7%.

Furthermore, effects which appear small in mixed cell populations can be masking large effects in sub-populations. Whilst we attempted to identify cell-type independent changes by correction for major cell-type composition, this does not rule out the possibility of an apparently small global effect being driven by a small ageing-related sub-population with larger effect sizes - especially given the documented propensity for tissue-specific expression patterns of tRNA genes^{9,10}.

We have rewritten the Abstract significantly in accordance with the word limitations and removed 'strongly directional' as requested above.

Finally, we have included this additional sentence in the Discussion:

Whilst the changes in DNA methylation we have observed are relatively small (i.e. 3.7% between 4-year to 78-year pools in tRNA-iMet-CAT-1-4), this is consistent with the effect size seen in the majority of age-DMPs in many other studies^{1,2}, except for the noted extreme outliers, such as EVOLV2¹¹. Furthermore, effect size cut-offs are observed to remove a large fraction of true age-DMPs³.

A Significant Age Related Hypermethylation Percentage of significant windows by feature type

B CpG Density Permutation Analysis

In Fig. 6 the authors present WGBS data of CpG methylation across tRNAs in newborn, adult and centenarian. How do the authors explain that there are no significant differences between newborn and adult, albeit aging-associated DNA methylation generally follows a logarithmic trajectory in childhood?

We agree with the reviewer that the lack of a significant difference between the neonate and the 26-year-old is deserving of further discussion. Regarding this observation, the logarithmic rate of change observed in DNA methylation in childhood has been clearly documented in the sites employed in DNA methylation clocks¹² and seen in other array-based studies. Optimal clock sites are selected because they exhibit changes in state throughout the lifespan to produce the best predictive values across the entire age range. Whilst these aging-associated DNA methylation changes follows a logarithmic trajectory in childhood, it should be noted even from array data, other ageing signals at particular loci have been observed, such as a gain in variability with age¹. The trajectory of tRNA gene set changes may be revealing differences in the specifics of the manifestations of ageing at different points in the lifecourse. These modifications in tRNA gene loci may represent biological changes that are more

gradual over the lifetime and lack the logarithmic shifts in the young, or are only prevalent in the later years. Recent ageing biology research has highlighted specific differences in aged humans, such as changes in patterns of chromatin openness in CD8 T cells¹³. This was represented in aged naive cells by a loss of promoter chromatin accessibility, due to decrease binding of the methylation-sensitive transcription factor NRF1, with associated reduced ability to transcribe respiratory chain genes. Also, other recent findings in aged individuals include a significant expansion of cytotoxic CD4 T cells in supercentenarians compared with less 'healthy' younger individuals¹⁴.

We have made the following addition to the Discussion section.

We observed a predominantly unmethylated state across fetal (Figure S7) and adult tissues (Figure S6) at tRNA gene loci, consistent with the high rate of transcription at many tRNA gene loci. We suspect that the tRNA genes largely remain unmethylated through development and that the moderate increases in DNAm that we are observing with age at these loci are being driven by changes arising primarily in older individuals. Distinct biological changes have been observed recently in aged individuals^{13,14}. Our data are supportive of the tRNA gene set representing a distinct functional unit, which lacks the logarithm change in the young observed strongly in CpGs employed in 'clocks'¹². This would also be consistent with the lack of significant differences in the tRNA loci detected between the neonate and the 26-year-old adult in the Heyn *et al.*⁸ data. This low baseline DNA methylation of the tRNA genes may also explain why we predominantly observe hypermethylation with age. Whether this is driven by mechanisms, such as aberrant DNA methylation targeting of the tRNA loci or specific sub-celltype effects with age, will require further experimental investigation.

- 3) In line 147 the authors mention their "novel pooling approach" of a total of 190 blood samples in four age categories. It is not entirely clear what is novel about this approach and the results would be more convincing by independent targeted bisulfite sequencing of individual samples.

We agree with the reviewer that independent targeted bisulfite sequencing of individuals would definitely provide increased power. However, it would be vastly more expensive both in term of sequencing costs and source material (DNA). The 'novelty' to which we were referring to with our pooling approach lies in the economy of this method as it permits testing of larger numbers with significantly lower sequencing costs and relatively smaller amounts of DNA. We have added the following line, to the manuscript to clarify the nature of our application of this pooling approach but have also removed the term 'novel' to reduce confusion.

This approach permitted us to assay tRNA gene DNA methylation across a large number of individuals whilst requiring a minimal amount of DNA from each individual (80-100ng), and costing ~1/24th as much as performing sequencing individually... Therefore, this confirmed the utility of our ~~this novel~~ pooling approach.

- 4) Apparently several tsRNAs reveal moderate expression changes – yet, with no evident overlap related to DNA methylation. How heterogeneous is the expression level within a given sample (particularly in the small subset of cells where DNA methylation is gained)? Would differential expression of tsRNAs impact on the aging process? These aspects might at least need further discussion.

Yes, we agree with the reviewer that the interpretation of these results are complex. Subsequently, in accordance with the recommendations of Reviewer 2, we have now removed this section pertaining to possible changes in tRNA / tsRNA expression due to the difficulty in accurately interpreting these data.

We have removed the results section entitled: **Expression of tRNAs in Blood with Age**, and the corresponding methods section entitled: **Assaying tRNA expression in blood with MINTmap**

5) The manuscript comprises 14 figures which are sometimes difficult to read. Some of them might be combined into multi panel figures, better labeled (e.g. TRUE and FALSE is not intuitive), color codes should be explained (e.g. Fig 6), and some aspects might be moved to the supplement to make the figures more reader friendly (e.g. the ideograms in Fig 9).

We agree with the reviewer and have consolidated several figures into panels, moved one to the supplement (Circos figure, formerly Figure 5) and removed another (former Figure 2)

Consolidating figures 3 & 4 into a single panel See response to point 1

Consolidating figure 6 into a single panel, with a new addition

Consolidating figures 11 & 12 into a single panel

Consolidating figures 7, 8 & 9 into a single panel

Reviewer #2

1) The abstract was 3 times the maximum length for this journal. I normally do not comment on article length, but because so much editing will be required to get this into a format that is

publishable, it puts the burden on reviewers to sort through it all, and it will require a fair amount of revision before final consideration for publication in this forum.

Abstract (Word Limit = 150)

The epigenome deteriorates with age, potentially impacting on ageing-related disease.

Here, we interrogate the DNA methylation state of the genomic loci of human tRNA. Whilst arising from only ~46kb (<0.002% genome), this information transfer machinery is the second most abundant cellular transcript. tRNAs also control metabolic processes known to affect ageing, through core translational and additional regulatory roles.

We identified a genomic enrichment for age-related DNA hypermethylation at tRNA loci. Analysis in 4,350 MeDIP-seq peripheral-blood DNA methylomes (16-82 years), classified 44 and 21 hypermethylating specific tRNAs at study- and genome-wide significance, respectively, contrasting with 0 hypomethylating. Validation and replication (450k array & independent targeted Bisphite-sequencing) supported the hypermethylation of this functional unit. The strongest consistent signals, also independent of major cell-type change, occur in tRNA-iMet-CAT-1-4 and tRNA-Ser-AGA-2-6.

This study is this first comprehensive evaluation of the genomic DNA methylation state of human tRNA genes and reveals a discreet hypermethylation with advancing age.

2) I generally appreciated both the introduction and discussion sections, which hit on important parts of prior published work and the implications of this work. The Results however haven't been carefully edited, and it seems to be a "throw everything on the wall and see what sticks" draft. For a tRNA researcher who cares about the specific genes being discussed, it was frustrating not having the complete list of tRNAs being discussed, beyond the 2-3 that were found to be consistently significant after cell-type correction. In the beginning of the Results section (line 89) "We identified 21 genome-wide significant and 44 study-wide significant results" ... please give a summary table of the 21 and 44 tRNA genes, by name, and the different data sets that do or do not support them. Far too often, P-values were given, and gene counts given, but no actual reference to a complete table with the full set of genes mentioned, which is what at least half your audience will really care about. Simply referring to a summary Figure 5 (in Results, line 96), without specific gene names, is not very helpful.

We thank the reviewer for these useful comments to aid improving the Results section of our manuscript. We have now extended Table 1 to include the 16 blood cell-type corrected study-wide significant tRNAs from the MeDIP-seq model, as well as the results for these loci from the 450k array and independent Targeted BiS-seq analyses. Furthermore, as requested we have now included extended Supplementary Tables (Supplementary Files S1-S4) with the complete tRNA results for the total 44 study-wide tRNAs for MeDIP-seq, including their Targeted Bisulfite Sequencing, 450k arrays, and mouse RRBS age analysis results, as relevant.

Extended Table 1 - Now Figure 4

tRNA	Window	MeDIP		450k Array		Targeted BiS-seq	
		Slope	p-value	Slope	p-value	Slope	p-value
tRNA-Gln-CTG-7-1	Chr1:147,800,750-147,801,250	0.84	2.60e-05				
tRNA-Glu-TTC-1-1	Chr2:131,094,500-131,095,000	1.11	4.64e-09				
	Chr2:131,094,250-131,094,750	1.00	1.12e-07				
	Chr2:131,094,750-131,095,250	1.00	3.28e-07				
tRNA-His-GTG-1-2	Chr1:146,544,500-146,545,000	0.92	1.38e-06				
tRNA-His-GTG-2-1	Chr1:149,155,750-149,156,250	1.05	2.98e-08				
	Chr1:149,155,500-149,156,000	0.83	1.37e-05				
tRNA-Ile-AAT-10-1	Chr6: 27,251,500- 27,252,000	1.07	1.45e-08			1.30	1.22e-03
	Chr6: 27,251,750- 27,252,250	0.90	1.86e-06			1.30	1.22e-03
tRNA-Ile-AAT-4-1	Chr17: 8,130,000- 8,130,500	1.19	2.98e-10	19.63	8.92e-06	-0.74	6.88e-04
	Chr17: 8,130,250- 8,130,750	0.77	3.99e-05	19.63	8.92e-06	-0.74	6.88e-04
tRNA-Ile-TAT-2-2	Chr6: 26,987,750- 26,988,250	0.97	7.25e-07	4.16	1.17e-02	-0.60	3.84e-01
tRNA-iMet-CAT-1-4	Chr6: 26,330,500- 26,331,000	1.28	2.83e-11	13.01	6.07e-06	4.54	9.35e-04
	Chr6: 26,330,250- 26,330,750	1.13	2.89e-09	13.01	6.07e-06	4.54	9.35e-04
tRNA-Leu-TAG-2-1	Chr14: 21,093,250- 21,093,750	1.04	9.38e-08			2.49	8.77e-03
	Chr14: 21,093,500- 21,094,000	0.94	8.50e-07			2.49	8.77e-03
tRNA-Pro-AGG-2-2	Chr6: 26,555,500- 26,556,000	1.04	3.97e-08				
	Chr6: 26,555,250- 26,555,750	1.01	9.58e-08				
tRNA-Ser-ACT-1-1	Chr6: 27,261,250- 27,261,750	0.97	3.53e-07			0.66	1.45e-01
tRNA-Ser-AGA-2-6	Chr17: 8,129,750- 8,130,250	1.21	1.16e-10	20.87	6.72e-05	0.62	4.28e-02
	Chr17: 8,130,000- 8,130,500	1.19	3.03e-10	20.87	6.72e-05	0.62	4.28e-02
tRNA-Ser-TGA-2-1	Chr6: 27,513,000- 27,513,500	0.90	3.58e-06	87.21	1.38e-04	-0.25	5.74e-01
tRNA-Val-AAC-1-2	Chr5:180,590,750-180,591,250	0.91	3.28e-06				
tRNA-Val-AAC-4-1	Chr6: 27,648,500- 27,649,000	1.07	1.25e-08	40.06	9.90e-03		
	Chr6: 27,648,750- 27,649,250	0.95	4.31e-07	40.06	9.90e-03		
tRNA-Val-CAC-2-1	Chr6: 27,247,750- 27,248,250	0.85	2.33e-05	59.16	5.05e-06		

- 📄 Supplementary File S1
- 📄 Supplementary File S2
- 📄 Supplementary File S3
- 📄 Supplementary File S4

3) The authors have been appropriately conservative in their assessment of the significance of DNAm changes. What is surprising, though, is the particular tRNA genes that appear to be most significantly affected. Two of the three most significant genes are located within the 3' UTR of the CTC1 gene, only 300 bp apart, which puts them in the same epigenetic context as a third tRNA gene, Thr-AGT-1-2, only 300 bp away from the other CTC1-overlapping genes. I am concerned that the methylation dynamics of all three of these tRNAs are influenced by the overlapping CTC1 gene – which does not necessarily cast doubt on the results, but is surprising and not what I would have expected. Why isn't Thr-AGT-1-2 also in the same significance bin as the other two? The third significant tRNA, iMet-CAT-1-4, is only about 1kb away from Trp-CCA-3-2, and appears to be in the same epigenetic neighborhood, but is not found to be significant. Again, this makes me curious why these genes are not getting similar significance as those extremely nearby, and should be addressed or acknowledged by the authors.

We concur with the reviewer a more detailed examination is warranted of the immediate genomic context of the results especially for tRNA genes in close genomic proximity. We have further examined this locus and have now included a new figure in the manuscript (Figure 3) which we hope will clarify this (also see below). Whilst the NCBI RefSeq annotation for CTC1 (NM_025099.6) extends over all three of the neighboring tRNAs, the Ensembl/HAVANA merge transcript annotation in GENCODE 19 only covers tRNA-Ile-AAT-4-1 with a secondary transcript (CTC1-002, ENST00000449476.2) which is subject to nonsense mediated decay. The new figure shows the 3kb up and down stream of tRNA-iMet-CAT-1-4 and of tRNA-Ser-AGA-2-6, (also covering tRNA-Ile-AAT-4-1). The windowing approach of the MeDIP-seq data analysis is clearly delineated to illustrate how it is that tRNA-iMet-CAT-1-4 (Fig 3A), for example, has distinct results to tRNA-Trp-CCA-3-2.

Legend: MeDIP-seq results (blood cell-type corrected model) for **A**) tRNA-iMet-CAT-1-4 as well as **B**) tRNA-Ser-AGA-2-6 and tRNA-Ile-AAT-4-1 exhibiting ageing-related DNA hypermethylation. Top to Bottom: Chromosome ideogram; Gene locations (GENCODE 19); MeDIP-seq DNA methylation ageing -log₁₀ p-value results (shown for each 500 bp overlapping window) ; MeDIP-seq slope of change with age; MeDIP-seq Medium coverage (Reads per Millions, RPM) calculated across all samples; CpG density (%); CpG locations (red lines); 500bp overlapping windows. The sharp peaks suggest that the results are localised to individual tRNA genes not the entire genic locus. One window overlapping tRNA-Ile-AAT-4-1 also partially overlaps tRNA-Ser-AGA-2-6. The 3' UTR of the CTC1 transcript CTC1-002 (ENST00000449476.2, GENCODE 19), which is subject to nonsense mediate decay, overlaps tRNA-Ile-AAT-4-1. tRNA genes with similar CpG density are exhibiting differing age-related DNAm change patterns.

4) The authors don't do a particularly good job of explaining or contextualizing why some loci are being hypermethylated, and others hypomethylated, and what the physiological implications of two very different changes over time might have on the aging phenotype. For example, there was limited discussion or interpretation of the puzzling results of Ile-AAT-4-1 having a negative BiS-seq slope. Again, with Ile-AAT-4-1 is just 300bp from Ser-AGA-2-6, why are the BiS-seq

slopes so different? Glossing over this (apparently) contradictory data is very troubling and will draw questions from anyone reading this carefully.

We agree with the reviewer that we did not adequately discuss the contradictory results for tRNA-Ile-AAT-4-1. As discussed above (Reviewer 1; Point 2), the predominant change that can be observed being DNA hypermethylation is partly a consequence of the generalized hypomethylation of these loci in development/early life. This is consistent with the fact that they are highly expressed loci. Regarding the inconsistency of the tRNA-Ile-AAT-4-1 result - this is covered by a MeDIP-seq 500bp window which exhibited genome-wide significant hypermethylation, but also partially overlaps tRNA-Ser-AGA-2-6 (as seen in Figure 3B). Whilst the 450k array probe overlapping tRNA-Ile-AAT-4-1 (cg06382303) appears to replicate this hypermethylation, it is a borderline case for exclusion flagged by Zhou et al.¹⁵ due to non-unique 30bp 3' subsequence. In the targeted Bisulfite-sequencing data, tRNA-Ile-AAT-4-1 actually exhibited a loss of methylation. Therefore, this may suggest that the hypermethylation signal observed at this locus in the MeDIP-seq data could have been 'pulled up' by the neighbouring tRNA-Ser-AGA-2-6 hypermethylation signal.

Along with the new figure detailed above, we have also made the following addition to the Discussion:

Regarding the inconsistent tRNA-Ile-AAT-4-1 result, it is covered by a MeDIP-seq 500bp window which exhibited genome-wide significant hypermethylation, but also partially overlaps tRNA-Ser-AGA-2-6 (Figure 3B). Whilst the 450k array probe overlapping tRNA-Ile-AAT-4-1 (cg06382303) appears to replicate this hypermethylation, it is a borderline case for exclusion flagged by Zhou et al.¹⁵ due to non-unique 30bp 3' subsequence. In the targeted Bisulfite sequencing data, tRNA-Ile-AAT-4-1 exhibited a loss of methylation. Therefore, this may suggest that the hypermethylation signal observed at this locus in the MeDIP-seq data could have been 'pulled up' by the neighbouring tRNA-Ser-AGA-2-6 hypermethylation signal. Of the 16 study-wide significant tRNA genes, only these two of these have a shared significant window, furthermore, in the expanded set of 44 only tRNA-Thr-AGT-1-2 could be similarly affected.

- 5) As mentioned earlier, the authors don't focus as much as I would hope / expect on the genomic neighbourhoods of the tRNA genes this study has found, which seems like a missed opportunity to interpret why these tRNAs, and not others that are in equally "highly transcribed" regions, are so affected. Two of the three overlap a critically important and interesting CTC1 gene, and there is no further mention of this?(!) By focusing on just the 3 most significant tRNA genes for much of the downstream analyses, perhaps some broader understanding is being missed by not looking at patterns among the broader 21 and 44-tRNA sets. As long as the authors are consistently clearly which tRNAs are significant in which statistical framework, it is fine to mention and discuss the broader sets, given that these might become more significant with broader studies in the future. For example, what is the proximity of the 21/44 tRNAs to protein coding or other tRNA genes? As someone who studies human and other mammalian tRNA genes in great detail, I can't understand why just those tRNA genes rose to significance and not many others that show tissue or cellular response specificity. I do not question the conservative methylation results, but feel as though this is only a very partial picture of what is happening in aging because from a tRNA biology perspective, it doesn't give readers any leads on why these loci are strongly affected and not others.

We agree with the reviewer's assessment that we have not done enough to provide the genomic context for our results and exam potential highly transcribed regions. We set out to explore this further by clustering the tRNA gene set into local groupings to directly address the reviewer's question

about genomic neighbourhoods. We examined the distribution of the age-hypermethylating tRNAs among those clusters. However, consistent with the discussion above (point 3) illustrating the narrow peak nature of those significant DNA methylation age changes over the tRNA genes, our findings do not support the occurrence of larger regional clusters of DNA methylation change. We have made the following addition to the Results section:

tRNA-iMet-CAT-1-4 is located in the largest tRNA gene cluster in the human genome at chr6p22.2-1. This cluster contains 157 tRNA genes spanning the 2.67Mb from tRNA-iMet-CAT-1-2 to tRNA-Leu-AAG-3-1, and also hosts a histone gene microcluster.

...

To place these hypermethylating tRNA genes in their genomic context we examined how the extended set of 44 non-blood cell-type corrected study-wide significant tRNAs clustered with other tRNA genes. We grouped together all tRNAs within 5Mb of one another and then required that a cluster contain at least 5 tRNA genes with a density of at least 5 tRNA genes per Mb (see Methods). This yielded 12 major tRNA gene clusters containing a total of 353 tRNA genes, including 42 of the 44 non-blood cell corrected study-wide significant tRNA genes (Figure 4d). The hypermethylating tRNA genes were identified to be spread evenly among these clusters proportionately to their size (% ageing tRNA per cluster; non-significant one-way ANOVA).

In reference to the reviewer point about our lack of elaboration on the overlap with CTC1 we have made the following additions:

Results

tRNA-Ile-AAT-4-1 and tRNA-Ser-AGA-2-6 are neighbours and are located on chromosome 17 (Figure 3B). Notably tRNA-Ile-AAT-4-1 and tRNA-Ser-AGA-2-6 have a third close neighbour tRNA-Thr-AGT-1-2 which does not show significant age-related hypermethylation. We observe a similar pattern of sharp peaks of significance closely localised around the other loci in the study-wide significant set. GENCODE 19 places tRNA-Ile-AAT-4-1 in the 3'UTR of a Nonsense-mediated decay transcript of *CTC1* (CTC1-002, ENST00000449476.2) and not of its primary transcript. Going forward we refer to these most robustly corrected sets of 3 and 16 tRNA genes as the genome-wide and study-wide significant tRNA genes respectively.

Discussion

The location of tRNA-Ser-AGA-2-6 and tRNA-Ile-AAT-4-1 immediately downstream of *CTC1* and of tRNA-Ile-AAT-4-1 within the 3'UTR of an alternate isoform of *CTC1*, which undergoes nonsense mediated decay raises the possibility that the gene body epigenetic regulation of *CTC1* may impact on the state of these tRNA genes. *CTC1*'s function in telomere maintenance¹⁶, DNA replication licensing¹⁷, and its role in a rare progeroid condition¹⁸ indicate that it has ageing-relevant biology. The possible relationship between the regulation of *CTC1* and that of the tRNA genes downstream of it warrants further study.

- 6) While the methylation results were intriguing and generally convincing, the attempt to assess transcription, and relate that to gene loci was extremely weak. There is an immense amount of (much uncharacterized) processing which creates tRNA-derived small RNAs from full length tRNAs, and it is becoming clear that full-length tRNA abundance is not reflected in fragment abundance in most cases. While I applaud the heroic attempts to convert MINTbase transcription data to something insightful, it is ultimately not scientifically convincing. Angiogenin and a host of RNases create tRNA fragments in response to cellular conditions, and

some tRNA fragments are stabilized (and accumulate) thanks to interactions with other proteins in the cell. Thus, aside from the “many to many” fragment to full-length tRNA to tRNA locus mapping gauntlet, the data presented in this section are off-topic and probably wrong (you *must* use full-length tRNA sequencing methods like DM-tRNA-seq to have any chance to assess full-length tRNA abundance of specific transcripts). I recommend that this part be dropped from the manuscript because it dilutes the scientific rigor of earlier conclusions. Similarly, trying to use tRNA fragment data to assess pre-tRNA abundance, and to estimate full-length tRNA just doesn't work (we've tried with much better data).

We thank the reviewer for their positive comments regarding our attempt to explore these MINTbase transcriptional data for changes with age, especially at the loci at which we identified DNA methylation changes. We also agree with the reviewer's conclusion regarding the complexity and considerable difficulty of this analysis. Consequently, due to its limitations and as advised by the reviewer, we have now removed this section.

We have removed the results section entitled: **Expression of tRNAs in Blood with Age**, and the corresponding methods section entitled: **Assaying tRNA expression in blood with MINTmap**

7) The brief section on mouse tRNA epigenetics seemed half-hearted. Do any of the mouse genes identified correlate with the syntenic or orthologous human genes that are in the 3/21/44 gene sets? A recent publication that has a map of tRNA gene synteny might be helpful (PMID: 31857444). If not, can you better explain or relate your mouse findings to the results with human tRNA genes? Right now, this section seems incomplete and does not meet the same standards as the earlier work.

We thank the reviewer for referring us to this excellent relevant paper from Thornlow et al.¹⁹ to improve our mouse tRNA analysis. We made use of this synteny information to compare our mouse results more directly with their human orthologs. Unfortunately, this was not able to add to our conclusions, but we have still included this additional exploration in the manuscript in the Results section as below.

In order to investigate the mouse results further we made use of data from Thornlow et al.¹⁹, which had established tRNA ortholog sets for 29 mammalian species. They identified 197 mouse tRNAs as having direct human orthologs with 44 of these included in the mouse results from Petkovich et al. However, unfortunately, although 2 of the top 3 tRNAs (tRNA-Ser-AGA-2-6 & tRNA-Ile-AAT-4-1) have mouse orthologs (tRNA-Ser-AGA-2-2 & tRNA-Ile-AAT-1-1), they are not covered in these mouse data. Furthermore, none of the tRNAs showing significant hypermethylation in mice (at nominal $p < 0.05$) had human orthologs in our MeDIP-seq study-wide significant hypermethylating set. Therefore, whilst we cannot directly compare specific tRNA loci due to this lack of coverage, it is interesting that the small number of significant tRNA genes in the mouse data also hypermethylate with age.

8) It is entirely up to the authors, but using the term “tRNAome” frequently (while, yes, it is mentioned in other papers) is about as elegant as referring to all protein genes as the “mRNAome”. I'd recommend simply “tRNA gene set” or “all tDNA” in its place, but that is just a suggestion.

We have remove the term 'tRNAome' from the manuscript and depending on the context replaced it with either 'tRNA genes' or 'tRNA gene set'.

- 9) There are major errors in Figure 2, where the gene count numbers are completely wrong in many cases (e.g., Arg and Asn tRNA counts have flipped to the wrong anticodons — Asn GTT and Ala AGC are the most common tRNA genes, yet they are listed as single copy). Also, the available information about high-confidence tRNA genes was not used, which we see as particularly relevant here. In a manuscript with way too many figures, I'd recommend just dropping it because there are no insights drawn by essentially an (inaccurate) re-publication of GtRNAdb information. Also, gene counts alone are extremely misleading, and scientifically mean almost nothing without an understanding of which loci are active and which are not (see PMID: 31857444).

The reviewer is correct, we erroneously inverted the colour scale that indicates tRNA gene count in previous Figure 2. We thank the reviewer for their close attention to detail in noticing this. This was a figure intended mainly to help orientate the epigenomic/genomic audience for this paper that may not be as intimately familiar with the human tRNA genes. We thank the reviewer for this feedback and have concluded that this figure is not adequately fulfilling its intended purpose and have removed it.

Removed Figure 2

- 10) Figure 5 has way too much information to be absorbed, and is very difficult to get a “summary” impression from — just too much data squished down into too little room. For important comparisons between data sets, I would represent in simplified comparisons and possibly moving this to the supplement unless it clearly illustrates key points in the manuscript (it only seems to be referred to once in the main text).

We agree with the reviewer and have moved this figure to the Supplement (now Supplementary Figure S9) and also extended Table 1 (see above point 2) to capture some of the key comparisons captured in this figure with the remaining points covered in the new supplemental data files (Supplementary Data Files S1-4).

- 11) Results, line 314: Ref #29 is cited, but a much more current and thorough analysis of tRNA gene mutation rate can be found in Thornlow et al., 2018 (PMID: 30127029).

We thank the reviewer for bringing our attention to this paper. The reference initially used here²⁰ remains relevant for the insight into the population prevalence of genetic variants in the tRNA genes so we have retained it in addition to the reviewer's suggestion. We made the following addition to the manuscript in response to this comment:

Despite strong purifying selection maintaining **very** low variation in tRNA gene bodies, **tRNA genes are subject to high levels of transcription-associated mutagenesis (TAM) leading to elevated mutation rates over evolutionary time in their immediate flanking sequences²¹.**

- 12) Figure 11: Epilogos are cited, but perhaps the contribution of ChromHMM, from which these states were actually computed, should also be cited here (ref #98 in manuscript).

We thank the reviewer for pointing out this oversight and we have added a reference to the Ernst et al. 2011²² paper regarding the ChromHMM segmentation algorithm.

We further explored the activity of the tRNAome in public Chromatin segmentation data in blood (Epilogos Blood & T-cells set)^{22,23}. (In Caption): Chromatin segmentation data from the Epilogos^{22,23} 'Blood & T-cell' 15 State model (tRNA genes +/- 200bp).

13) Abstract, line 10: Please use current numbers available at the GtRNAdb — this number is very out-dated.

Yes, we agree with the reviewer and have now updated our manuscript to reflect the current number of tRNA genes in the high confidence set and also qualified any uses of the expanded set of 'tRNA-like' sequences. We included these later loci to see if any DNAm changes we observed were common to related homologous and potentially paralogous sequences including pseudogenes to aid in identifying any sequence specific effects. The full list of tRNA features used is provided as a BED file in the supplementary materials as Supplementary File S5.

We have modified these numbers in the Abstract, Introduction and Results

Introduction When liberally considering the 610 loci annotated as tRNA genes, tRNA pseudo genes, nuclear encoded mitochondrial tRNA genes and possibly some closely related repetitive sequences, these features (gtRNAdb²⁴) cover <46 kb (including introns) which represents <0.002% of the human genome²⁰. Yet a subset of these genes produce the second most abundant RNA species next to ribosomal RNA²⁵ and are required for the production of all proteins.

Results There are 416 high confidence tRNA genes in the human genome, we initially examined an expanded set of 610 tRNA-like features identified by tRNAscan including pseudogenes, possible repetitive elements, and nuclear encoded mitochondrial tRNAs^{24,26}.

14) General comment on all Figure legends: it is helpful to state, either in the title of the figure / table, or in the text, what is the scientific point that is most important about the figure — just labeling "Heatmap of X and Y..." is not very helpful to the reader, but "Heatmap showing three tRNAs are significant across tissues" is helpful in guiding the reader through complex figures of from large data sets. Please carefully select the main text figures that illustrate key points, and put the rest / full data set figures in the supplement.

We thank the Reviewer for this feedback and have revised our figure legend titles to better convey the key point being made by our figures. For example, below is our revised legend for the heatmap legend cited by the reviewer:

Figures 11 & 12 have been consolidated into a single panel Figure 7 and legend revised:

Figure 7: A) Heatmap²⁷ of Mean DNA methylation of tRNA sorted Blood Cell Types (data from Reinus et al., probes covering 8 of the 16 study-wide significant tRNA in 7 cell-type fractions from 6 males via GSE35069²⁸). This illustrates the ability of tRNA gene methylation to separate the myeloid from the lymphoid lineage and the higher methylation in the lymphoid lineage of the 3 cell-type corrected genome-wide significant tRNAs (iMet-CAT-1-4, Ser-AGA-2-6 & Ile-AAT-4-1). B). Chromatin Segmentation data for 'Blood & T-cell' from Epilogos 15 State model^{22,23}. Proportion represents the frequency of the predominant chromatin state at a given tRNA (+/- 200bp) for 14/16 study-wide significant tRNAs covered compared to other 371 available tRNAs. The ageing-related hypermethylated tRNAs are enriched for enhancer-like chromatin state (Fisher's Exact p = 0.01).

15) What would be the impact on tRNA abundance and translation, if any, if these are multi-copy genes?

We agree that the reviewer's question is an important one that needs addressing. Even if changes in mature tRNA expression at individual loci were compensated for by the multicopy nature of tRNA genes, the role of tRNA derived small RNAs may still be impacted. We have made the following addition to the Discussion to make this point:

tsRNA abundance has been associated with locus specific tRNA gene expression, in some cases independent of mature tRNA levels²⁹. This has important implications for the interpretation of our results given the multi-copy nature of genes like tRNA-iMet-CAT-1-4, as even if expression levels of mature iMet tRNAs are unaffected by changes in one copy's DNA methylation, these changes could still influence the levels of particular tsRNAs derived from specific tRNA loci.

Additionally, recent results from Gerber et al.³⁰ show a mechanism by which polII can function in the regulation of certain polIII transcribed loci including several tRNA genes. This may have implications for the interpretation of our results as the effects of DNA methylation on the regulation of polII may be relevant to the transcriptional activity of tRNA genes. We added the following line to the Discussion concerning this new result:

Furthermore, recent data from Gerber et al.³⁰ indicates Pol-II may also have a dynamic regulatory role in tRNA expression.

References

1. Sliker, R. C. *et al.* Age-related accrual of methylomic variability is linked to fundamental ageing mechanisms. *Genome Biol.* **17**, 191 (2016).
2. Xu, Z. & Taylor, J. A. Genome-wide age-related DNA methylation changes in blood and other tissues relate to histone modification, expression and cancer. *Carcinogenesis* **35**, 356–364 (2014).
3. Zhu, T., Zheng, S. C., Paul, D. S., Horvath, S. & Teschendorff, A. E. Cell and tissue type independent age-associated DNA methylation changes are not rare but common. *Aging (Albany, NY)*. **10**, 3541–3557 (2018).
4. Horvath, S. DNA methylation age of human tissues and cell types. *Genome Biol.* **14**, R115 (2013).
5. Field, A. E. *et al.* DNA Methylation Clocks in Aging: Categories, Causes, and Consequences. *Mol. Cell* **71**, 882–895 (2018).
6. Robinson, M. D., Statham, A. L., Speed, T. P. & Clark, S. J. Protocol matters: which methylome are you actually studying? *Epigenomics* **2**, 587–598 (2010).
7. Su, D. *et al.* Distinct Epigenetic Effects of Tobacco Smoking in Whole Blood and among Leukocyte Subtypes. *PLoS One* **11**, e0166486 (2016).

8. Heyn, H. *et al.* Distinct DNA methylomes of newborns and centenarians. *Proc. Natl. Acad. Sci.* **109**, 10522–10527 (2012).
9. Sagi, D. *et al.* Tissue- and Time-Specific Expression of Otherwise Identical tRNA Genes. *PLOS Genet.* **12**, e1006264 (2016).
10. Dittmar, K. A., Goodenbour, J. M. & Pan, T. Tissue-specific differences in human transfer RNA expression. *PLoS Genet.* **2**, e221 (2006).
11. Slieker, R. C., Relton, C. L., Gaunt, T. R., Slagboom, P. E. & Heijmans, B. T. Age-related DNA methylation changes are tissue-specific with ELOVL2 promoter methylation as exception. *Epigenetics and Chromatin* **11**, 1–11 (2018).
12. Bell, C. G. *et al.* DNA methylation aging clocks: challenges and recommendations. *Genome Biol.* **20**, 249 (2019).
13. Moskowitz, D. M. *et al.* Epigenomics of human CD8 T cell differentiation and aging. *Sci. Immunol.* **2**, eaag0192 (2017).
14. Hashimoto, K. *et al.* Single-cell transcriptomics reveals expansion of cytotoxic CD4 T cells in supercentenarians. *Proc. Natl. Acad. Sci. U. S. A.* **116**, 24242–24251 (2019).
15. Zhou, W., Laird, P. W. & Shen, H. Comprehensive characterization, annotation and innovative use of Infinium DNA methylation BeadChip probes. *Nucleic Acids Res.* **45**, gkw967 (2016).
16. Gu, P. *et al.* CTC1-STN1 coordinates G- and C-strand synthesis to regulate telomere length. *Aging Cell* **17**, e12783 (2018).
17. Wang, Y., Brady, K. S., Caiello, B. P., Ackerson, S. M. & Stewart, J. A. Human CST suppresses origin licensing and promotes AND-1/Ctf4 chromatin association. *Life Sci. Alliance* **2**, e201800270 (2019).
18. Sargolzaeiaval, F. *et al.* CTC1 mutations in a Brazilian family with progeroid features and recurrent bone fractures. *Mol. Genet. Genomic Med.* **6**, 1148–1156 (2018).
19. Thornlow, B. P. *et al.* Predicting transfer RNA gene activity from sequence and genome context. *Genome Res.* **30**, 85–94 (2020).
20. Parisien, M., Wang, X. & Pan, T. Diversity of human tRNA genes from the 1000-genomes project. *RNA Biol.* **10**, 1853–1867 (2013).
21. Thornlow, B. P. *et al.* Transfer RNA genes experience exceptionally elevated mutation rates. *Proc. Natl. Acad. Sci.* **115**, 8996–9001 (2018).
22. Ernst, J. *et al.* Mapping and analysis of chromatin state dynamics in nine human cell types. *Nature* **473**, 43–49 (2011).
23. Meuleman, W. epilogos. (2019). Available at: <https://epilogos.altius.org/>.

24. Chan, P. P. & Lowe, T. M. GtRNADB: a database of transfer RNA genes detected in genomic sequence. *Nucleic Acids Res.* **37**, D93-7 (2009).
25. Lodish, H. *et al.* *Molecular Cell Biology, 4th edition.* (W. H. Freeman, 2000).
26. Lowe, T. M. & Chan, P. P. tRNAscan-SE On-line: integrating search and context for analysis of transfer RNA genes. *Nucleic Acids Res.* **44**, W54–W57 (2016).
27. Gu, Z., Eils, R. & Schlesner, M. Complex heatmaps reveal patterns and correlations in multidimensional genomic data. *Bioinformatics* **32**, 2847–2849 (2016).
28. Reinius, L. E. *et al.* Differential DNA Methylation in Purified Human Blood Cells: Implications for Cell Lineage and Studies on Disease Susceptibility. *PLoS One* **7**, e41361 (2012).
29. Torres, A. G., Reina, O., Stephan-Otto Attolini, C. & Ribas de Pouplana, L. Differential expression of human tRNA genes drives the abundance of tRNA-derived fragments. *Proc. Natl. Acad. Sci.* **116**, 201821120 (2019).
30. Gerber, A., Ito, K., Chu, C.-S. & Roeder, R. G. Gene-Specific Control of tRNA Expression by RNA Polymerase II. *Mol. Cell* **78**, 765-778.e7 (2020).

REVIEWERS' COMMENTS

Reviewer #1 (Remarks to the Author):

The authors have now addressed each of my comments and tempered several claims. They hardly performed new experiments and removed some sections (e.g. on the expression of tRNAs in blood with age). It remains unclear if the moderate DNA methylation changes are functionally relevant and how they are affected by the cellular composition. Several figures have now been combined, but the formatting might be further adjusted (e.g. the font size in Fig. 5). Overall, I feel that the paper is scientifically sound and it will be of interest for researchers with interest in aging-related DNA methylation changes.

Reviewer #3 (Remarks to the Author):

The authors described significant associations between tRNA methylation level and ageing. The findings maybe novel, but the methods which used to obtain the finding maybe flawed. This is mainly to do with the initial analysis of the twin data also questionable pooling techniques.

1. Linear regression was used with initial discovery dataset which contain a large amount of data from twin studies. Because of the nature of twins, the data should be correlated, thus linear regression may not be the ideal model to use. Models such as mixed effect model may be better.
2. Both of the original twin dataset and the BiS-seq data set were highly biased toward female. I wonder if there is any need to include the male samples. Maybe the authors can show that there is no difference in term of methylation between male and female. The BiS-seq dataset was pooled, this may or may not work, I would personally encountered positive and negative experience with pooling sequencing. Also one pool is entirely of pregnant women, I applaud the authors for mentioning this, but it does put into the question of whether methylation alters due to pregnancy.

3. The discovery dataset (Twins dataset) were originally quantile normalized with RPM. When the authors extract tRNA information, what happens with a tRNA resides on the boundary of two bins?

4. Figure 1 really should be a table instead. In figure 1, are all p value from linear regressions? If these are the p values from regression, then effect size (beta) should be shown also. Why are some cell without data in Figure 1 for 450K array and BiS-Seq. For 450K, I imagine that was due to no coverage in the probes. For BiS-seq did author filtered them out because of depth < 25 as stated in the method section? Depth ≥ 25 is rather a strange cutoff, as the majority of the studies use 20.

5. Figure S8 does not demonstrate the enrichment of tRNAs efficiently. I would suggest the authors produce an IGV or USSC genome browser track and show entire genome methylation level then mark the tRNA regions.

6. How permutation analysis for enrichment is not clearly explained. "A random set of 500bp windows" how big is a set?

7. Also regarding enrichment analysis, the authors also conducted a Fisher exact test with sample size 3001, why not 4350? Also please give the 2x2 table used for Fisher's exact test.

Reviewer #4 (Remarks to the Author):

I was asked to evaluate the manuscript and whether the authors have adequately addressed the Reviewer #2's comments from a previous review. I will focus my review on this matter. Reviewer 2 was very thorough in their review and I have no additional concerns regarding the manuscript

that have not been brought up by other reviewers.

Comment 1: The reviewers have adequately shortened the abstract.

Comment 2: In my opinion, the added tables (now Figure 4 and supplementary Files 1-4) significantly increase the value of the manuscript, and will be highly useful to the audience. The comment was adequately addressed.

Comment 3: The author added a figure which answers the reviewers' questions. I find that the labels in the figure are too small, and font size should be increased (and made black instead of grey). But the figure clearly and very nicely illustrates the loci of the tRNAs in question and confirms that the hypermethylation is specific to the tRNA genes and not the overall locus. This also plays into the answer to question four.

Comment 4: The authors adequately address reviewer 2's question, but should add a sentence that for the 2 loci the data should thus be regarded as inconclusive (which doesn't take away anything from the manuscript).

Comment 5: The authors have added a supporting figure (discussed in comment 3), which shows that the methylation is - despite the localization of the tRNA gene near protein coding sequences - specific to the tRNA locus. They added an additional analysis and clustered the tRNAs that are encoded in close proximity but found that hypermethylation is spread evenly among clusters. The question how the localization of 3 tRNAs in close proximity to the CTC1 gene affects the regulation of tRNA methylation is beyond the scope of the manuscript, but is addressed in the discussion.

Comment 6: This comment recommends the removal of a section and the authors have done so.

Comment 7: The authors have attempted the comparison of their dataset with a published dataset as suggested by the reviewer, but very little overlap was observed. This is fine.

Comments 8 -14: The authors have followed the recommendation of the reviewer and edited the manuscript accordingly.

Comment 15: The authors have added some additional discussion to satisfy the reviewer's question. The discussion is appropriate, as this is mainly a matter of speculation, but will be an interesting topic of future research.

Overall the concerns have in my opinion been adequately addressed, with a very small revision mentioned above in comments 3 and 4.

Responses to Second Round of Reviewers' Comments

We thank the reviewers for their time and comments on our manuscript.

Reviewer #1 (Remarks to the Author):

The authors have now addressed each of my comments and tempered several claims. They hardly performed new experiments and removed some sections (e.g. on the expression of tRNAs in blood with age). It remains unclear if the moderate DNA methylation changes are functionally relevant and how they are affected by the cellular composition. Several figures have now been combined, but the formatting might be further adjusted (e.g. the font size in Fig. 5). Overall, I feel that the paper is scientifically sound and it will be of interest for researchers with interest in ageing-related DNA methylation changes.

We thank the reviewer for their confirmation that the paper is scientifically sound, as well as positive comments that we have adequately addressed their points and tempered specified claims.

As recommended by the reviewer, we have adjusted the formatting of figures including the font of Figure 5 in the latest version of the manuscript.

To further highlight the pan-tissue nature of our findings, in response to the reviewer's additional comment above on cellular composition, we have now also included the normal adult versus normal fetal tissue results from the public TCGA data across a range of tissue types in the 2 array-covered top tRNAs highlighted in the Abstract (iMet-CAT-1-4 and Ser-AGA-2-6) – please see Figure 7C. The following additional text have been added to the Results:

“These two tRNAs do show broad age-related hypermethylation across a range of tissues in a comparison between fetal to adult, with interestingly, directionally consistent but higher levels for iMet-CAT1-4 in the adrenal gland (Fig 7C)”

Reviewer #3 (Remarks to the Author):

The authors described significant associations between tRNA methylation level and ageing. The findings maybe novel, but the methods which used to obtain the finding maybe flawed. This is mainly to do with the initial analysis of the twin data also questionable pooling techniques.

1. Linear regression was used with initial discovery dataset which contain a large amount of data from twin studies. Because of the nature of twins, the data should be correlated, thus linear regression may not be the ideal model to use. Models such as mixed effect model may be better.

We agree with the reviewer that a mixed effects model including twin pair as a random effect would be a suitable method to account for the correlation within our samples due to genetic relatedness. The approach that we took to address this important issue was to directly query this in subset analyses, rather than by statistically estimating this. We performed analyses on subsets of the data containing only individuals without their twins, or separately only themselves at different time points (i.e., longitudinal) in these sets. This showed consistent age-related DNA hypermethylation findings (please see Results: – line 91 in Body text). However, as requested by the reviewer, we have also rerun the analysis as a linear mixed effect model similar to previous analyses (Roos et al., 2016; Xu & Taylor, 2014) with batch, lymphocyte, monocyte, neutrophil and eosinophil cell count data, as fixed effects, and family id and zygosity as random effects. This also showed a very similar directional tRNA loci results to the linear regression. Of the windows significant at $p < 0.05$ in this model 76 were hypermethylating versus 9 hypomethylating with age. The complete results

for this model are included in Supplementary file 1. The windows overlapping tRNA-Ser-AGA-2-6 and tRNA-iMet-CAT-1-4 still rank highly (6th and 17th, respectively) in statistical significance in this analysis. The result from this model, therefore, also strongly display age-related tRNA loci hypermethylation.

We had added this additional text to the Results Section:

We also ran a number of additional analyses to investigate this directional observation further, including performed linear mixed modelling with batch, lymphocyte, monocyte, neutrophil and eosinophil cell count data, as fixed effects, and family id and zygosity as random effects. Thus, we have observed the same consistent hypermethylation trend with age across a wide array of models, with and without correction for cell-type composition, and when either correcting for structure in our sample population or when examining those sub-populations separately.

2. Both of the original twin dataset and the BiS-seq data set were highly biased toward female. I wonder if there is any need to include the male samples. Maybe the authors can show that there is no difference in term of methylation between male and female. The BiS-seq dataset was pooled, this may or may not work, I would personally encountered positive and negative experience with pooling sequencing. Also one pool is entirely of pregnant women, I applaud the authors for mentioning this, but it does put into the question of whether methylation alters due to pregnancy.

Yes, the reviewer raises an excellent point that the addition of a small proportion of males may unduly make the dataset more heterogeneous. There are identifiable differences between male and female DNA methylation patterns with age, with males generally showing slightly greater age acceleration with age than women excepting some tissues e.g. breast (Horvath, 2013, p. 20; Horvath et al., 2016). However, both show broadly similar pattern of age-related DNA methylation changes. In fact, in our previous ageing-related analysis of this dataset within GWAS common-disease associated loci, revealed that for these strong MeDIP DMRs sex was not a significant factor. In that analysis, we performed a female-only analysis in the discovery stage and mixed sex in the replication set (Bell et al., 2016). Strong replication was seen in our results between both sets. In addition, it should be noted that we focus exclusively on the autosomes in all these analyses, so differences in DNA methylation on the sex chromosomes were not an issue in this work. Furthermore, we also chose a completely female set for our targeted bisulfite sequencing, except for one of the 4-year-old pools, to match the heavily female bias of our discovery set (see Table 1). At that 4-year age point there were insufficient female samples, so were split into two groups by sex. There was no discernible difference between these two pools in their Horvath clock estimate (see Figure 5C), even though the rate of change in DNAm is known to be steep in the early years (Alisch et al., 2012). However, as suggested by the reviewer we have rerun the MeDIP-seq analysis in female and male only, which in female samples showed very similar results to the combined model. tRNA-iMet-CAT-1-4 ranks 3rd and tRNA-Ser-AGA-2-6 4th by p-value and 258 windows hypermethylate and 22 hypomethylate at the p-value threshold of <0.05 in the batch and blood cell-type corrected model. Only 10 male samples have complete blood cell-type haematological data, so even whilst these results are very underpowered, the same trend is seen with almost twice the windows hypermethylating versus hypomethylating (48 versus 26) at the p-value threshold of 0.05 in the batch and blood cell-type corrected model. Full model results for male and female only models have also been added to Supplementary file 1.

Whilst we cannot rule out an effect related to pregnancy in the 28-year all female group, there is no compelling evidence these are driving a major trajectory shift in the age-related DNA methylation sufficient in magnitude to impact on the results of our study. In fact, our conclusions would still stand even removing these young adult pools, due our findings in the other age-group pools. Whilst it is almost certain subtle signatures of DNAm change exist during pregnancy due to endocrine and physiological changes, the current

literature is conflicting, probably due to the resolution of methodology (Chen et al., 2017; Gruzieva et al., 2019). One might speculate that the presence of the minute, but detectable cell-free DNA derived from the fetus circulating in the maternal blood supply might produce a slight bias towards underestimating maternal DNA methylation age signatures. The Horvath age prediction for these pregnancy 28-year pools does not deviate significantly from the trend line, giving an expected value between the 4 year and 63 year time points. Finally, we acknowledge with the reviewer that pooling approaches can lead to a loss of power. However, we confirmed the fact that these pooled DNA methylation data when used for epigenetic analysis produced a highly accurate Horvath clock age estimation of the pool ages, and, furthermore, worked consistently with our replication/validation. This implies our observed age-related DNAm results were strong enough to overcome any potential power reduction.

3. The discovery dataset (Twins dataset) were originally quantile normalized with RPM. When the authors extract tRNA information, what happens with a tRNA resides on the boundary of two bins?

Yes, we agree this needs more clarification. We considered all the bins overlapping the tRNAs. Due to the overlapping sliding window nature of the DNA methylome (500bp semi-overlapping windows, See Figure 3), all genomic regions are covered by 2 windows. Thus, because of the size of tRNA (all <200 bp), all the individual tRNAs were covered by 2-3 windows. tRNAs were considered significant if any window overlapping them showed significant changes. We have clarified this with additional text in the Methods section

“Thus, all individual tRNA genes are covered by 2-3 windows. tRNAs were considered significant if any window overlapping them showed significant changes.”

4. Figure 1 really should be a table instead. In figure 1, are all p value from linear regressions? If these are the p values from regression, then effect size (beta) should be shown also. Why are some cell without data in Figure 1 for 450K array and BiS-Seq. For 450K, I imagine that was due to no coverage in the probes. For BiS-seq did author filtered them out because of depth < 25 as stated in the method section? Depth ≥ 25 is rather a strange cutoff, as the majority of the studies use 20.

Unfortunately, including colour coding in a table, as we have, designates it as a figure. Yes, p-values are from the linear regressions and the slope (beta) for the linear model results from the normalised RPM data is shown in the column prior in this table. Not all of these loci were targeted for bisulfite sequencing and some which were did not pass our QC procedures. Therefore, the table simply shows those sites for which data was available, this applies to both the 450k array and bisulfite sequencing datasets. We have also adjusted the array slope value to be in the same scale (% methylated 0-100) as the targeted bisulfite sequencing data slopes as opposed to the beta value (proportion methylated 0-1) used in the array to make these more easily comparable.

With regards to the choice of a sequencing depth cut-off, as stated in the ‘Targeted Bisulfite Sequencing’ Methods subsection, employing this slightly more stringent threshold of 25 reads robustly reduced our off-target results. Also, whilst we do acknowledge 20 is common threshold in studies, 25 has also been implemented by others (Komori et al., 2015).

5. Figure S8 does not demonstrate the enrichment of tRNAs efficiently. I would suggest the authors produce an IGV or USSC genome browser track and show entire genome methylation level then mark the tRNA regions.

We agree with the reviewer that this is an excellent idea to display the results and have now included UCSC genome browser bigwig tracks (hg19).

This is available through trackhub at <http://epigenome.soton.ac.uk/TrackHub/hub.txt>

Tracks show, in non-overlapping windows: mean, median and variance in RPM across all samples used in cell-type corrected analysis (n = 3,001); the percentage of samples with an exactly 0 RPM score in a given window (useful for spotting technical issues); the slope and $-\log_{10}(p\text{-values})$ for batch corrected, and blood cell-type corrected Age models

6. How permutation analysis for enrichment is not clearly explained. "A random set of 500bp windows" how big is a set?

We apologise if this was not clear to the reviewer. The size of the set varied with the number of windows overlapping the features being assessed. Thus, for tRNAs it is the number of windows overlapping tRNA windows (1,320), for Bivalent chromatin (42,841), and for targets of the polycomb group proteins (13,502). We have added a line to the Methods section to clarify this:

The number of windows selected in each permutation for these features was for tRNAs 1,320; for bivalent chromatin 42,841; and for polycomb group protein target promoters 13,502.

7. Also regarding enrichment analysis, the authors also conducted a Fisher exact test with sample size 3001, why not 4350? Also please give the 2x2 table used for Fisher's exact test.

As the hypermethylation results were so strong in the entire dataset, and although the cell-specific results are fascinating in their own right, we focused here on the smaller number of potential pan-cellular changes. Therefore, we used the set of 3,001 as this is the number of samples that possessed haematological laboratory blood cell-type count information ('TwinsUK MeDIP-seq methylomes' in Methods). We performed the permutation and proportional enrichment analyses using this refined blood cell-type corrected model, hence the 3,001 samples. The Fisher's test was performed on the number of windows significant in the linear modelling over the tRNA gene set compared the number of significant windows in the entire genome. Similarly, for other features e.g. Exons, CGIs etc. Thus, the contingency tables are derived from the number of significant windows over a feature (nSig, at the same fixed tRNA study wide significance threshold of 8.36×10^{-5}), the number of windows over that feature (nWins) and these same values subtracted from the whole genome counts. We have included all this information in a new Supplementary data file: SF2_prop_enrich_fishers.xlsx.

Reviewer #4 (Remarks to the Author):

I was asked to evaluate the manuscript and whether the authors have adequately addressed the Reviewer #2's comments from a previous review. I will focus my review on this matter. Reviewer 2 was very thorough in their review and I have no additional concerns regarding the manuscript that have not been brought up by other reviewers.

Overall the concerns have in my opinion been adequately addressed, with a very small revision mentioned above in comments 3 and 4.

We thank the reviewer for evaluating the previous comments from Reviewer 2 on our manuscript and have focused on comments 3 and 4 as they have indicated.

Comment 3: The author added a figure which answers the reviewers questions. I find that the labels in the figure are too small, and font size should be increased (and made black instead of grey). But the figure clearly and very nicely illustrates the loci of the tRNAs in question and confirms that the hypermethylation is specific to the tRNA genes and not the overall locus. This also plays into the answer to question four.

As requested, we have altered this Figure 3 to have larger labels and black font.

Comment 4: The authors adequately address reviewer 2's question, but should add a sentence that for the 2 loci the data should thus be regarded as inconclusive (which doesn't take away anything from the manuscript).

We agree with the reviewer that the data are inconclusive for tRNA-Ile-AAT-4-1. However, we consider the results are solid for tRNA-Ser-AGA-2-6, as it replicated strongly in the targeted bisulfite sequencing and validated well in the 450k array results, which was not the case for tRNA-Ile-AAT-4-1

As such, we have added the following line to the Discussion:

These factors considered together suggest that the result for tRNA-Ile-AAT-4-1 should be regarded as inconclusive. (p 22 line 307)

References

- Alisch, R.S., Barwick, B.G., Chopra, P., Myrick, L.K., Satten, G.A., Conneely, K.N., Warren, S.T., 2012. Age-associated DNA methylation in pediatric populations. *Genome Res.* 22, 623–632. <https://doi.org/10.1101/gr.125187.111>
- Bell, C.G., Xia, Y., Yuan, W., Gao, F., Ward, K., Roos, L., Mangino, M., Hysi, P.G., Bell, J., Wang, J., Spector, T.D., 2016. Novel regional age-associated DNA methylation changes within human common disease-associated loci. *Genome Biol.* 17, 193. <https://doi.org/10.1186/s13059-016-1051-8>
- Chen, S., Mukherjee, N., Janjanam, V.D., Arshad, S.H., Kurukulaaratchy, R.J., Holloway, J.W., Zhang, H., Karmaus, W., 2017. Consistency and variability of DNA methylation in women during puberty, young adulthood, and pregnancy. *Genet. Epigenetics* 9. <https://doi.org/10.1177/1179237X17721540>
- Gruzieva, O., Merid, S.K., Chen, S., Mukherjee, N., Hedman, A.M., Almqvist, C., Andolf, E., Jiang, Y., Kere, J., Scheynius, A., Söderhäll, C., Ullemar, V., Karmaus, W., Melén, E., Arshad, S.H., Pershagen, G., 2019. DNA Methylation Trajectories During Pregnancy. *Epigenetics Insights* 12. <https://doi.org/10.1177/2516865719867090>
- Horvath, S., 2013. DNA methylation age of human tissues and cell types. *Genome Biol.* 14, R115. <https://doi.org/10.1186/gb-2013-14-10-r115>
- Horvath, S., Gurven, M., Levine, M.E., Trumble, B.C., Kaplan, H., Allayee, H., Ritz, B.R., Chen, B., Lu, A.T., Rickabaugh, T.M., Jamieson, B.D., Sun, D., Li, S., Chen, W., Quintana-Murci, L., Fagny, M., Kobor, M.S., Tsao, P.S., Reiner, A.P., Edlefsen, K.L., Absher, D., Assimes, T.L., 2016. An epigenetic clock analysis of race/ethnicity, sex, and coronary heart disease. *Genome Biol.* 17, 171. <https://doi.org/10.1186/s13059-016-1030-0>
- Komori, H.K., Hart, T., LaMere, S.A., Chew, P.V., Salomon, D.R., 2015. Defining CD4 T cell memory by the epigenetic landscape of CpG DNA methylation. *J. Immunol. Baltim. Md* 1950 194, 1565–1579. <https://doi.org/10.4049/jimmunol.1401162>
- Roos, L., van Dongen, J., Bell, C. G., Burri, A., Deloukas, P., Boomsma, D. I., ... Bell, J. T. (2016). Integrative DNA methylome analysis of pan-cancer biomarkers in cancer discordant monozygotic twin-pairs. *Clinical Epigenetics*, 8(1), 7. <https://doi.org/10.1186/s13148-016-0172-y>
- Xu, Z., & Taylor, J. A. (2014). Genome-wide age-related DNA methylation changes in blood and other tissues relate to histone modification, expression and cancer. *Carcinogenesis*, 35(2), 356–364. <https://doi.org/10.1093/carcin/bgt391>